# Tropospheric ozone retrieval by a combination of TROPOMI/S5P measurements with BASCOE assimilated data

Heue Klaus-Peter[1,2], Loyola Diego[1], Romahn Fabian[1], Zimmer Walter[1], Chabrillat Simon[3], Errera Quentin[3], Ziemke Jerry[4], and Kramarova Natalya[4]

[1]Institut für Methodik der Fernerkundung am Deutschen Zentrum für Luft- und Raumfahrt (DLR), Oberpfaffenhofen, Germany
[2]Technische Universität München (TUM), Munich, Germany
[3]Royal Belgian Institute for Space Aeronomy (BIRA-IASB), Brussels, Belgium
[4]NASA Goddard Space Flight Center (GSFC), Greenbelt, Maryland, USA

**Correspondence:** Klaus-Peter Heue: Klaus-Peter.Heue@DLR.de

**Abstract.** We present a new tropospheric ozone data set based on TROPOMI/Sentinel-5 Precursor (S5P) total ozone measurements combined with stratospheric ozone data from the Belgian Assimilation System for Chemical ObsErvations (BASCOE) constrained by assimilating ozone observations from the Microwave Limb Sounder (MLS). The BASCOE stratospheric data is interpolated to the S5P observations and subtracted from the TROPOMI total ozone data. The difference equals the tropospheric ozone residual column from the surface up to the tropopause. The tropospheric ozone columns are retrieved at the full spatial resolution of the TROPOMI sensor (5.5 x 3.5 km$^2$) with daily global coverage.

Compared to the OMPS-MERRA-2 data a global mean positive bias of 3.3 DU is found for the analysed period April 2018 to June 2020. A small negative bias of about -0.91 DU is observed in the tropics relative to the operational TROPOMI tropical tropospheric data based on the CCD algorithm through out the same period. The new tropospheric ozone data (S5P-BASCOE) is compared to a set of globally distributed ozone sondes data integrated up to the tropopause level. We found 2254 comparisons with cloud free TROPOMI observations within 25 km around the stations. In the global mean S5P-BASCOE deviates by 2.6 DU from the integrated ozone sondes. Depending on the latitude the S5P-BASCOE deviate from the sondes and between -4.8 and 7.9 DU, indicating a good agreement. However, some exceptional larger positive deviation up to 12 DU are found especially in the northern polar regions (north of 70 $^\circ$). The monthly mean tropospheric column as well as time series for selected places showed the expected spatial and temporal pattern, like the wave one structure in the tropics or the seasonal cycle including a summer maximum in the mid-latitudes.

## 1 Introduction

Tropospheric ozone is an important pollutant because it affects human health and crop growth. Especially respiratory and cardiovascular symptoms increase with short term exposure to enhanced ozone concentration (e.g. Fleming et al., 2018). In the global mean tropospheric ozone is responsible for about 10 % loss of the wheat production (e.g. Avery et al., 2011; Ainsworth et al., 2012), depending on the region and the crop the loss may reach up to 25 %. In the troposphere ozone is produced

by photo-chemical processes converting primary pollutants such as NOx and VOCs or directly by lightning. Stratospheric intrusion is another import source of tropospheric ozone. Due to its long lifetime of 20 to 30 days (e.g. Wu et al. , 2007) ozone can be transported over large distances. Moreover, tropospheric ozone acts as a greenhouse gas ($0.40 \pm 0.20$ W·m$^{-2}$, IPCC, 2013) and is an important source of OH which controls the lifetime of many other atmospheric species.

Currently several approaches are used to derive tropospheric ozone from satellite observations. In the tropics the Convective Cloud Differential method (Ziemke et al., 1998) can be used. The TROPOMI/S5P (TROPOspheric Monitoring Instrument on Sentinel 5 Precursor) tropical tropospheric ozone (Heue et al. , 2016; Heue et al., 2021b) has been generated operationally since December 2018 based on the CCD. The vertical ozone column above deep convective clouds gives an estimate of the stratospheric ozone column. It is assumed that the stratospheric ozone column varies slowly in time and latitude but is longitudinal constant. The stratospheric background column is averaged for a certain reference region (Indian Ocean, Indonesia to the Pacific Ocean) and subtracted from the total column for cloud free observations.

Ziemke et al. (2006) presented a limb nadir matching approach based on the combination of nadir observations from OMI (Ozone Monitoring Instrument) and limb observations form the Microwave Limb Sounder (MLS), both on the NASA Aura satellite. The nadir viewing OMI observes the total column and MLS provides the ozone vertical distribution from 0.02 hPa down to the upper troposphere. To retrieve the stratospheric column the MLS ozone profile is assimilated to Modern-Era Retrospective analysis for Research and Applications-2 (MERRA-2, Gelaro et al., 2017) and integrated above the tropopause. Both datasets are gridded to the same grid (1° latitude x 1.25° longitude) and only data with less than 30 % cloud coverage are considered. In addition also the a version with a direct combination of OMI and MLS data is available (https://acd-ext. gsfc.nasa.gov/Data_services/cloud_slice/new_data.html, March 2022, Ziemke et al., 2006). Both instruments are installed on the same platform and observe the same air mass within 7 minutes delay. The product was further improved and the OMI measurements were continued by OMPS (Ozone Mapping and Profiler Suite) nadir observations (Section 3.1.2).

SCIAMACHY (Scanning Imaging Absorption Spectrometer for Atmospheric Chartography) on Envisat (2002-2012) was capable of observing both the total column in nadir and the stratospheric profile in limb geometry. Ebojie et al. (2014) published the latest update of the limb nadir matching data based on SCIAMACHY observation. The algorithm is in principle similar to the one used for OMI-MLS, except that both total column and stratospheric ozone profile are observed with the same instrument in the UV range. The limb observations are used to retrieve a stratospheric ozone profile, the profile was then integrated above the tropopause to calculate the stratospheric columns. The difference between the total column, retrieved from the SCIAMACHY nadir observation, and the stratospheric column results in the tropospheric residual.

Miles et al. (2015) used an optimal estimation method (Rodgers, 2000) to retrieve the profile information from GOME, SCIAMACHY, OMI and GOME-2 nadir observations. The different sensitivity of the instruments to ozone absorption in the Hartley band and in the Huggins band and the temperature dependency of the ozone absorption cross section are the key parameters to retrieve the ozone profile. The data were analysed within ESA's CCI project and are regularly updated for EU's Copernicus Climate Change Service (C3S). The same physical background is used by Smithsonian Astrophysical Observatory (SAO) algorithm (Huang et al., 2017) to derive ozone profiles below 60km with 2.5 km vertical resolution from

OMI observations. Since December 2021 the TROPOMI/S5P operational ozone profiles (Veefkind et al., 2021) are available, which also contain a tropospheric ozone subcolumns up to 6km and from 6-12 km.

The above mentioned ozone profiles or tropospheric data are mostly based on the ozone absorption in the UV (260 to 360 nm), except for MLS. The IASI (Infrared Atmospheric Sounding Interferometer) instruments on the MetOp (A, B and C) satellites make use of the infrared ozone absorptions between wavenumber 1025 and 1075 $cm^{-1}$. The FORLI (Fast Optimal Retrievals on Layers for IASI) algorithm (Boynard et al., 2018) is also based on an optimal estimation method and retrieves profiles of 39 layers up to 39 km altitude and one additional layer up to the top of atmosphere. The data are restricted to cloud coverage of less than 13 %.

A combined IASI+GOME-2 retrieval (Cuesta et al., 2013) enhances the sensitivity relative to GOME-2 or IASI especially for the lower troposphere, below 3 km. Both instruments are installed on the MetOp satellite series and collocated spectral observations are analysed simultaneously. The final data have the same spatial resolution as IASI.

The tropospheric ozone burden can be retrieved by assimilation of both the total ozone column and the stratospheric ozone profile using chemical transfer simulations. CAMS (Copernicus Atmosphere Monitoring Service) also uses $O_3$ total columns from TROPOMI and other satellite instruments to constrain the total ozone and MLS for the stratospheric column. Inness et al. (2019) showed that the additional assimilation of TROPOMI ozone columns improves the data quality in the tropical to mid-latitude troposphere. The CAMS ozone profiles can be downloaded at https://ads.atmosphere.copernicus.eu/cdsapp#!/ search?type=dataset (last access March 2022).

In this study we introduce a new tropospheric ozone dataset S5P-BASCOE, based on TROPOMI/S5P total ozone measurements and stratospheric ozone data provided by the Belgian Assimilation System for Chemical ObsErvations (BASCOE) constrained by MLS ozone profiles. The algorithm makes use of the high spatial resolution of the TROPOMI instrument (5.5 x 3.5 $km^2$). Sentinel 5P was launched in October 2017 and together with the future Sentinel-5 mission it will provide global measurements during the next decades. The BASCOE stratospheric ozone system provides a forecast, of stratospheric ozone profiles. In combination with the near-real-time (NRTI) S5P total ozone columns the tropospheric ozone column may also be provided in near real-time i.e. three hours after sensing.

In the next section 2 of the paper the tropospheric ozone retrieval is presented including a brief introduction of the total ozone column algorithm as well as the BASCOE assimilation. In the following section 3.1 the tropospheric ozone column data sets (OMPS-MERRA, S5P_CCD) and ozone sondes data will be explained briefly and comparisons relatie to these tropospheric ozone data sets are shown. Finally, tropospheric ozone results will be presented and briefly discussed.

## 2 Troposperic Ozone Retrieval

### 2.1 S5P-BASCOE

The S5P-BASCOE tropospheric ozone retrieval is based on a three step approach as described in the sections 2.2, 2.3 and 2.4. The main inputs and intermediate data are displayed in Figure 1. In a first step the total ozone column is retrieved from the TROPOMI/S5P observations (section 2.2), here we use the operational S5P NRTI products with a resolution of 7 x 3.5 $km^2$.

In the second step the ozone profiles from BASCOE (section 2.3) are integrated between the tropopause pressure and the top
of the atmosphere to obtain the stratospheric column in the resolution of 2.5° x 3.75° x 3 h in the last step the BASCOE
stratospheric column is interpolated in space and time to match the S5P observations and subtracted from the TROPOMI total
columns (section 2.4).

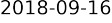

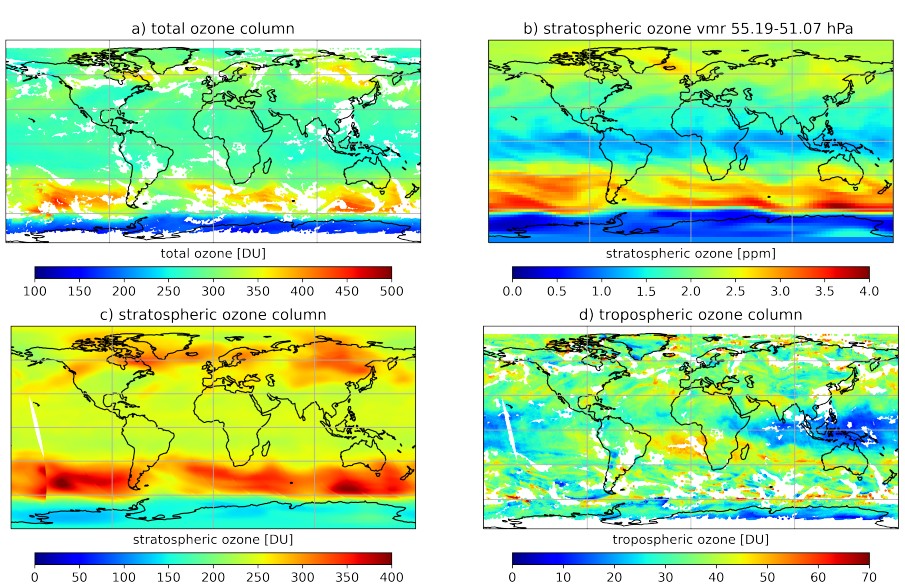

**Figure 1.** Overview of the tropospheric ozone retrieval:

a) TROPOMI NRTI total ozone column, white regions represent cloud screened or no data availability

b) BASCOE $O_3$ mixing ratio for 2018-09-16 12:00 UTC around 52 hPa, with a resolution of 2.5° by 3.75°.

c) Integrated stratospheric ozone column from BASCOE interpolated in space and time to the TROPOMI observation, the data have the
full S5P resolution, 7 x 3.5 km². At 170°W the first and the last of orbit of this day overlap, the time difference of $\approx$ 23 h between these
observations cause the jump in the stratospheric ozone columns.

d) Tropospheric ozone column calculated as the difference between total a) and stratospheric column c)

## 2.2   TROPOMI Total Ozone Retrieval

The Sentinel-5 Precursor (S5P) satellite was launched in October 2017 into a sun synchronous orbit with an equator crossing
time of 13:30. TROPOMI observes the atmosphere with a daily coverage and a spatial resolution of 5.5 x 3.5 km² (7 x 3.5
km² until 6th August 2019) and a spectral resolution of roughly 0.5 nm in the UV. The S5P near-real-time (NRTI) total ozone
product is based on the well established two step DOAS approach with an iterative Air Mass Factor (AMF) calculation (Loyola
et al., 2011; Hao et al., 2014). The slant column density is retrieved in the 325 to 335 nm wavelength range. The S5P cloud
algorithm provides cloud top height, cloud optical density and cloud fraction. The innovative approach in S5P is to treat clouds

as layers of scattering droplets (Loyola et al., 2018). Also in the ozone AMF calculations the same cloud model is applied (Heue et al., 2021a). Garane et al. (2019) showed that the NRTI total ozone column in general agrees well with ground-based observations but shows some bias in the polar to mid-latitude winter, which was caused by the albedo climatology (Kleipool et al., 2008) used in UPAS (Universal Processor for Atmospheric Spectrometers) version 1. To solve this problem in UPAS version 2 the surface albedo required for the AMF calculation is retrieved from the TROPOMI measurements using a full physics inverse machine learning method (Loyola et al., 2020). Currently the version 2.3.0 of UPAS is being used for generating the S5P NRTI total ozone product. Figure 1 a) shows an example of NRTI total column, the data are cloud filtered for further retrieval.

The presented tropospheric algorithm can be applied to the S5P vertical ozone columns retrieved with both NRTI and offline algorithm, as well as other satellites. In this paper we used total ozone products based on the UPAS version 2.1.3 of the NRTI algorithm and reprocessed internally at DLR.

### 2.3 BASCOE Assimilations of Ozone Profiles

This implementation of BASCOE was developed to improve the representation of stratospheric composition in the EU Copernicus Atmospheric Monitoring Service (CAMS), by providing independent analyses of ozone and five other species which are also observed by MLS (HCl, ClO, HNO3, $N_2O$, $H_2O$). These data are used to evaluate the analyses and forecasts of stratospheric ozone which are delivered operationally by CAMS (e.g. Sudarchikova et al., 2021) and also to verify research versions of the CAMS system where the stratospheric chemistry module from the BASCOE system is implemented into the CAMS system (Huijnen et al., 2016). Since it is an operational service, the BASCOE-FD (fast delivery) system has evolved with time due to the changes in the ECMWF (European Centre for Medium-Range Weather Forecasts) operational system. Moreover BASCOE-FD was adapted to the updates in MLS retrieval algorithm and those in the BASCOE system (see the changelog here: http://www.copernicus-stratosphere.eu/4_NRT_products/3_Models_changelogs/BASCOE.php, Dec. 2021). BASCOE-FD provides analyses of stratospheric ozone and other chemical species operationally with a timeliness of 3-5 days in order to allow the assimilation of the Aura-MLS offline dataset. Throughout the paper BASCOE and BASCOE-FD ares used as synonyms. The BASCOE-FD ozone fields are provided on a 2.5° latitude by 3.75° longitude grid with a temporal resolution of 3 hours. Since March 2016, BASCOE-FD uses a vertical grid with 86 levels from the surface to 0.01 hPa. An example of the BASCOE-FD ozone mixing ration for 2018-09-16 12:00 UTC between 55.19 and 51.07 hPa is shown in Figure 1 (b). During this period the Antarctic ozone hole was almost fully developed and the ozone mixing ratio above Antarctica was reduced .

An early version (q2.4) of BASCOE-FD has been evaluated against total ozone ground-based measurements, ozonesonde profiles and satellite profiles over the period 2009-2012 (Lefever et al., 2015). The agreement was usually within +/- 10 % but degraded to +/- 40 % in the tropical tropopause layer (TTL). The version used here (5.7) runs operationally since March 2016. It is evaluated every three months for the validation of the CAMS operational analyses, indicating stable biases which are usually smaller than 5 % in the middle stratosphere and 15 % in the TTL (e.g. Sudarchikova et al., 2021). In the upper stratosphere above 4 hPa pressure altitude, the BASCOE system has a small ozone deficit (Errera et al., 2019) which introduces a negative bias in the BASCOE-FD stratospheric ozone columns. This has been corrected using a time-latitude climatology of this bias

against MLS. This climatology is based on the BASCOE-FD analyses between July 2016 and March 2019 with a resolution of

5° latitude and 1 day. In this work the climatology was smoothed and linearly interpolated to 2.5° latitude. It varies between -1

and 4 DU (Figure 2). The integration of stratospheric columns from the BASCOE-FD analyses starts at dynamical tropopause

height as given in the BASCOE-FD data files. The calculation of the tropopause pressure is done in two independent steps,

first the PV, PT (potential vorticity, potential temperature) tropopause is calculated. Outside the tropics (outside 30° South

to North) the tropopause is defined as the Potential Vorticity isosurface at 3.5 PVU and inside the tropics as the isentropic

isosurface with a potential temperature of 380 K or 3.5 PVU, whatever is lower. The second step is based on the WMO (World

Meteorological Organisation) definition. The tropopause is given as the lowest altitude where the temperature lapse rate dT/dz

is less than 2 K/km and does not exceed 2 K/km in the next 2 km above. The potential vorticity and temperature are extracted

from ECMWF operational analyses at a reduced spatial resolution (T31) corresponding to the coarse grid of BASCOE-FD. In

the final step the two definitions are combined by choosing the lower altitude / higher pressure level. For practical reasons the

centre pressure level of the respective grid cell is given. The S5P-BASCOE data file also contains the corresponding tropopause

pressures. In addition, BASCOE-FD provides data with alternative tropopause definitions e.g. 2.5 PVU for the period from

August 2019 onward. The impact of the tropopause definition on the tropospheric ozone column is discussed in section 3.2.2.

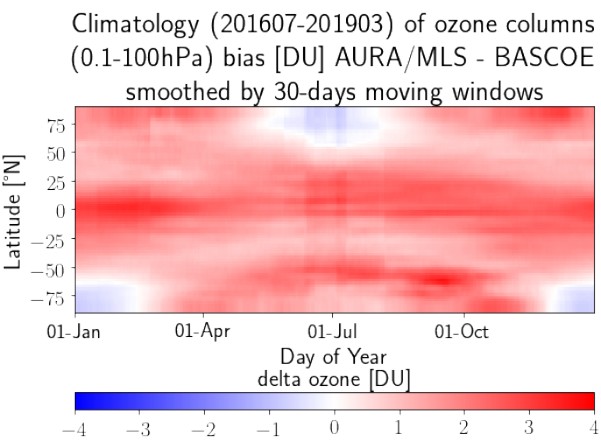

**Figure 2.** Time-latitude stratospheric ozone column bias climatology between MLS and BASCOE used to correct the BASCOE-FD strato-
spheric column.

## 2.4   S5P-BASCOE Tropospheric Ozone

The stratospheric ozone column is calculated from the BASCOE assimilated fields between the tropopause and the upper

lid i.e. 0.01 hPa. A correction term accounts for the BASCOE ozone deficit above 4 hPa. The latitude and time dependent

climatology is added to the stratospheric ozone column, the correction is in the order of 2 DU (Figure 2). The stratospheric

ozone column has the spatial and temporal resolution of 3.75° by 2.5° and 3 hours following the resolution of the BASCOE

ozone profile.

TROPOMI/S5P has a daily global coverage with a spatial resolution of 5.5 x 3.5 km$^2$. The BASCOE stratospheric ozone column is linearly interpolated in time and space to the TROPOMI pixel centre coordinate and observation time. Figure 1 (c) shows the interpolated stratospheric column including the ozone deficit correction (Figure 2). Some patterns are similar between the two subplot (1 (b) and (c)), however the interpolation and vertical integration also cause a significant smoothing. Moreover the stratospheric column is interpolated to the TROPOMI measurement time, note that the first and the last orbit in 1 (c) overlap over the Pacific Ocean but differ in time by 23 hours resulting in a discountious ozone column here.

Furthermore, the tropopause pressure as given in the BASCOE results is interpolated to the TROPOMI ground pixels, and stored with the tropospheric column. Clouds shield the lower tropospheric ozone measured by satellite UV-instruments. Because of that we only take TROPOMI observations with a cloud fraction of less than 20 % for computing the tropospheric ozone. In the final step the interpolated stratospheric column is subtracted from the total ozone column to compute the tropospheric residual, see Figure 1 (d).

## 3   Comparisons to tropospheric Data Sets

### 3.1   Other tropospheric Ozone Data

#### 3.1.1   TROPOMI_CCD

The tropical tropospheric ozone column based on the convective cloud differential (CCD) is an official TROPOMI product generated operational and regularly validated, the validation reports are available at https://mpc-vdaf.tropomi.eu/index.php/search (May 2022). The algorithm has been described in a previous publication (Heue et al. , 2016), and the S5P O3_TCL ATBD (algorithm theoretical basis document) (Heue et al., 2021b) therefore only a short summary is given here. In a reference region (70°E to 170°W) the above cloud column is calculated based on the TROPOMI OFFL (offline) total ozone column, based on GODFIT (GOME Direct FITting) algorithm version 4 as described in Lerot et al. (2010) and used in Inness et al. (2019). During the total column retrieval, a ghost column is added for the part of the ozone column that is shielded by clouds (inside or below the cloud). After subtracting the ghost column the remaining column equals the above cloud ozone.

The mean cloud altitude of the deep convective clouds in the reference regions is usually close to 10 km, according to the TROPOMI cloud retrieval (Argyrouli et al., 2021) but varies from cloud to cloud. To normalize the above cloud ozone column for the varying cloud altitudes to a reference level of 270 hPa, the partial ozone column between the cloud altitude and the reference level was added. This correction column is based on the climatology by McPeters and Labow  (2012). For clouds higher than 10 km we add a small correction to account for the ozone shielded between 10 km and the cloud, similarly for lower clouds a respective column is subtracted. Thereby the above cloud columns cover the same altitude range above 10 km . In the mean the correction term is low, because the mean cloud top height is close to the 10 km level. The cloud altitude corrected above cloud ozone column approximates the stratospheric column. However, the real tropical tropopause is well above 10 km or 270 hPa, the above cloud stratospheric approximation hence also includes the upper troposphere.

It is assumed that for each latitude band the stratospheric ozone column is constant along the longitude and varies only slowly in time and latitude. This assumption is in general used for the CCD algorithm and is only justified within the tropics, therefore the algorithm is limited to the latitude range between 20° S and 20° N. For several examples of BASCOE stratospheric column varied by less than 5 DU standard deviation within 6 days, along the longitude for 0.5° latitude. The temporal and spatial resolution is comparable to the S5P_CDD settings.

We subtract the stratospheric ozone column from the total ozone column for the cloud free observations (cloud fraction less than 10 %). The cloud free data are averaged within a certain latitude x longitude grid and a time period. Compared to the ESA's ozone CCI tropospheric ozone the spatial resolution was adapted to 0.5° latitude x 1° longitude. The temporal resolution for the CCI data set from GOME-2B and OMI is one month, with TROPOMI it is now reduced to 6 days for the stratospheric column and 3 days for the tropospheric column. Due to the latitudinal limitations of the CCD method the comparison between the two

S5P tropospheric ozone datasets can be performed only within the tropics. The different altitude ranges from the surface to 270 hPa for CCD or to the 380 K level ($\approx$ 80 to 130 hPa) for S5P-BASCOE causes a systematic difference. For the following comparison (section 3.2) we added the subcolumn between the 270 hPa reference level and the 380 K based on the ozone profile by McPeters and Labow (2012).

### 3.1.2   OMPS-MERRA-2 tropospheric Ozone

The evaluation of S5P-BASCOE tropospheric ozone includes comparisons with a research product of tropospheric column ozone derived by combining total column ozone from the Suomi National Polar orbiting Partnership (SNPP) Ozone Mapping Profiler Suite (OMPS) nadir-mapper (NM with stratospheric column ozone from Modern-Era Retrospective analysis for Research and Applications-2 (MERRA-2). Daily global maps of OMPS-MERRA-2 tropospheric column ozone were determined using a residual method similar to Ziemke et al. (2006) that subtracts stratospheric column ozone from total column ozone.

The OMPS-NM instrument measures total column ozone about three minutes from TROPOMI overpass, providing an ideal dataset for cross-comparisons with TROPOMI. Total ozone from OMPS is determined using a version 2.1 algorithm that includes aerosol adjustments and cloud optical centroid pressures (OCPs) retrieved from OMI. The algorithm is based on the well established TOMS V8 retrieval. More details on the retrieval and comparison to other datasets are discussed in McPeters et al. (2019). The OMPS data including quality evaluation are available from https://ozoneaq.gsfc.nasa.gov/data/omps/(Jan.

2022). The OMPS-NM provides full global coverage of the sunlit Earth each day, making 400 scans per orbit with 36 across-track measurements for each scan. OMPS field of view (FOV) is about 50 km by 50 km at nadir for the 300 to 380 nm band. The MERRA-2 data assimilation system (Gelaro et al., 2017) uses Aura OMI v8.5 total ozone and MLS v4.2 stratospheric ozone profiles to produce global synoptic maps of profile ozone from the surface to the top of the atmosphere; these profiles are reported every three hours (0, 3, ... 21 UTC) at a resolution of 0.625° longitude x 0.5° latitude. For each hourly map

and at each grid-point, MERRA-2 profile ozone was integrated vertically from the top of the atmosphere down to tropopause pressure to derive maps of stratospheric column ozone. Tropopause pressure was determined from MERRA-2 re-analyses using standard PV- Θ definition (2.5 PVU and 380 K). The resulting maps of stratospheric column ozone from MERRA-2 were then co-located and subtracted from OMPS total ozone, thus producing daily global maps of tropospheric column

ozone sampled at OMPS local time. These tropospheric ozone pixel measurements were binned to 1° latitude x 1° longitude resolution. In the following the dataset will be named OMPS-MERRA-2. MERRA-2 assimilated stratosphere column ozone was found to agree within ±2-3 DU with the collocated MLS measurements. Comparisons between collocated ozone sonde and OMPS-MERRA-2 tropospheric column ozone in the tropics and extra-tropics indicate mean differences varying from near zero to at most ≈ ±6 DU, and standard deviations from a few DU to at most ≈6-8 DU (Elshorbany et al., 2021). Largest differences and standard deviations were found in the mid- and high latitudes with smaller biases in the tropics. The OMPS-MERRA-2 tropospheric ozone columns were not filtered for clouds. There were no adjustments of any kind applied to the OMPS-MERRA-2 tropospheric column ozone.

In contrast to the CCD data, the OMPS-MERRA-2 tropospheric data provide a global data set for the comparison. However, the analysis approach is similar to the one presented here, in particular both use MLS data for quantifying the stratospheric contribution. Therefore, the following comparison is not based on fully independent data sets.

### 3.1.3 Ozone Sondes

Ozone sondes are regularly launched from various stations around the globe. The data are provided by the national services and can be downloaded via World Ozone and Ultraviolet Radiation Data Centre (https://woudc.org/data/explore.php, April. 2022). The sounding stations are globally distributed. For this comparison we considered sounding data from more than 60 stations. However, some stations launch a balloon every week while others once in a month. Also the spatial distribution is not uniform, while 9 stations in Europe provided roughly 800 soundings between 2018 and 2021 there was only one sounding station in China (Hong-Kong ≈ 70 soundings and three for the US mainland with about 220 profiles. The data distribution can be estimated from Figure 3.

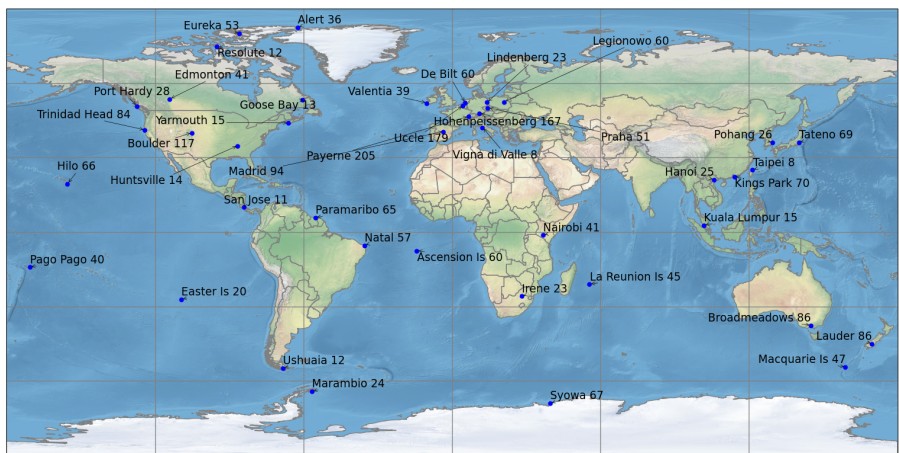

**Figure 3.** Global distribution of the ozone sounding stations, the number of soundings used in this study is given next to the station's name. Image wade using Natural Earth free vector and raster map data from naturalearthdata.com (Jan 2022).

**Table 1.** Mean difference and standard deviation of S5P-BASCOE relative to the individual datasets

| deviation | S5P_CCD | OMPS-MERRA-2 | sondes |
|-----------|---------|--------------|--------|
| DU | -0.91 $\pm$ 5.76 | 3.34 $\pm$ 7.64 | 2.6 $\pm$ 9.3 |
| % | -0.82 $\pm$21.71 | 14.59 $\pm$ 32.51 | 12.8 $\pm$ 29.1 |

## 3.2 Comparison Results

Our S5P-BASCOE tropospheric ozone data are compared to the data sets presented in section 3.1. For the S5P CCD (OFFL-O$_3$)
and S5P-BASCOE (NRTI-O$_3$) the total ozone columns are observed with the same instrument but are retrieved using different algorithms. OMPS-MERRA-2 and S5P-BASCOE share not only the similar retrieval approach but additionally BASCOE and MERRA-2 both assimilate MLS ozone profiles. Because of that the satellite-satellite comparisons are not based on fully independent measurements. The ozone sondes however are an independent and widely accepted validation data set. The results are described and discussed in the following sections a summary is given in table 1.

Both S5P_CCD and OMPS-MERRA-2 are gridded dataset with resolution of 0.5° x 1° and 1° x 1° latitude by longitude, respectively. The S5P-BASCOE data set is first gridded to 0.25°x0.25° for plotting and other applications, for the comparison we averaged the grids to match the resolution of S5P_CCD or OMPS-MLS.

### 3.2.1 Comparison to S5P_CCD

The tropical tropospheric ozone column retrieval (S5P_CCD) is described in section 3.1.1. The data are restricted to the in-
250 ner tropical range between 20° South and 20° North and include the vertical range up to 270 hPa. We use the same time period as for OMPS-MERRA-2 (sect. 3.2.2) from April 2018 to June 2020. The CCD data were quality filtered using a mini-mum qa-vaulue of 0.7 as recommended in Product Readme File (https://sentinels.copernicus.eu/documents/247904/3541451/ Sentinel-5P-OFFL-Tropospheric-Ozone-Product-Readme-File.pdf May 2022). The monthly plot for April 2018 (Figure 4) shows that the difference is mostly negative, S5P-BASCOE is lower than S5P_CCD, but the differences are typically less
than 5 DU. There is no systematic structure like land-sea bias to be found in the plots. The time series of the tropical aver-aged difference between S5P-BASCOE and S5P_CCD is illustrated in Figure 5. The figure also includes a comparison to the OMPS-MERRA-2 tropospheric column for both the tropical subset and the global scale which will be discussed in more detail in section 3.2.2. The daily averaged tropical differences to the CCD also show a negative bias of -0.91 DU and a standard vari-ation of 5.76 DU. Three smaller peaks are observed in June / July 2018 and a large one in May / June 2019. Some of the peaks
also occur in the comparison with OMPS-MERRA-2, especially in the tropical comparison. The large peak in May / June 2019 is not seen in the comparison to OMPS-MERRA-2 hence it results from a decrease in the S5P_CCD data in this period, the cause is not yet fully understood. Also for the first peak 10$^{th}$ June 2018 a deviation in the CCD data (not shown) contributes to the increase in the differences. For the next two peaks it seems to be an overestimation of the tropical ozone by S5P-BASCOE. The differences to the S5P_CCD also show a clear annual cycle, which is not seen relative to the OMPS-MERRA-2, neither for

the global nor for the tropical comparison. So most probably the S5P_CCD data cause the annual cycle in the difference, this can be confirmed by an annual cycle found in the differences to some sounding stations and GOME-2B_CCD as documented in the validation report (https://mpc-vdaf.tropomi.eu/index.php/search, May 2022).

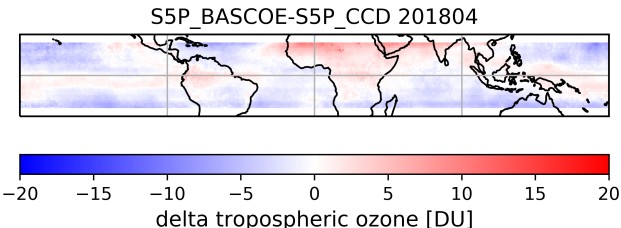

**Figure 4.** Monthly mean difference between S5P-BASCOE and S5P_CCD for April 2018.

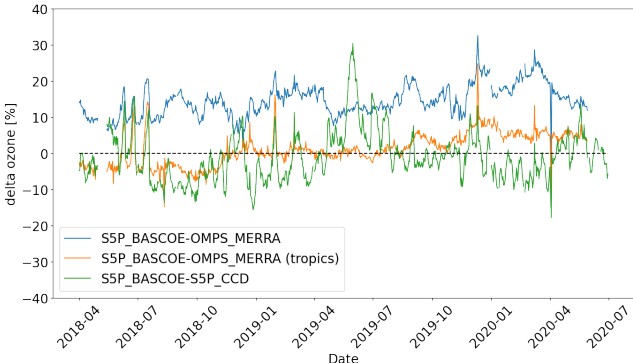

**Figure 5.** Time series of the differences between S5P-BASCOE against S5P_CCD (green), OMPS-MERRA-2 (blue), and the tropical subset of OMPS-MERRA-2 (orange). The CCD comparison focuses on the tropical region while for the OMPS-MERRA-2 also the global data sets were considered. In the temporal mean a negative bias ($\approx$0.91 DU or 0.81 %) is found relative to CCD data and a positive one (3.34 DU or 14.59 %) relative to OMPS-MERRA-2.

### 3.2.2 Comparison to OMPS-MERRA-2

Tropospheric ozone column retrieval from OMPS-MERRA-2 is described in section 3.1.2. The mean difference for May 2020
(Figure 6) shows an underestimation around 20° North especially over the Saharan dessert and an overestimation in the northern mid to high latitudes as well as in the southern high latitudes. This pattern is typical also for the other month included in this comparison exercise. In the mean an overestimation can be found, but for large parts of the world the differences are smaller than $\pm$5 DU.

The globally averaged difference between S5P-BASCOE and OMPS-MERRA-2 for each day is shown in Figure 5 together
with the differences to the CCD based tropospheric ozone. The differences vary between 2 and 6 DU with a mean difference

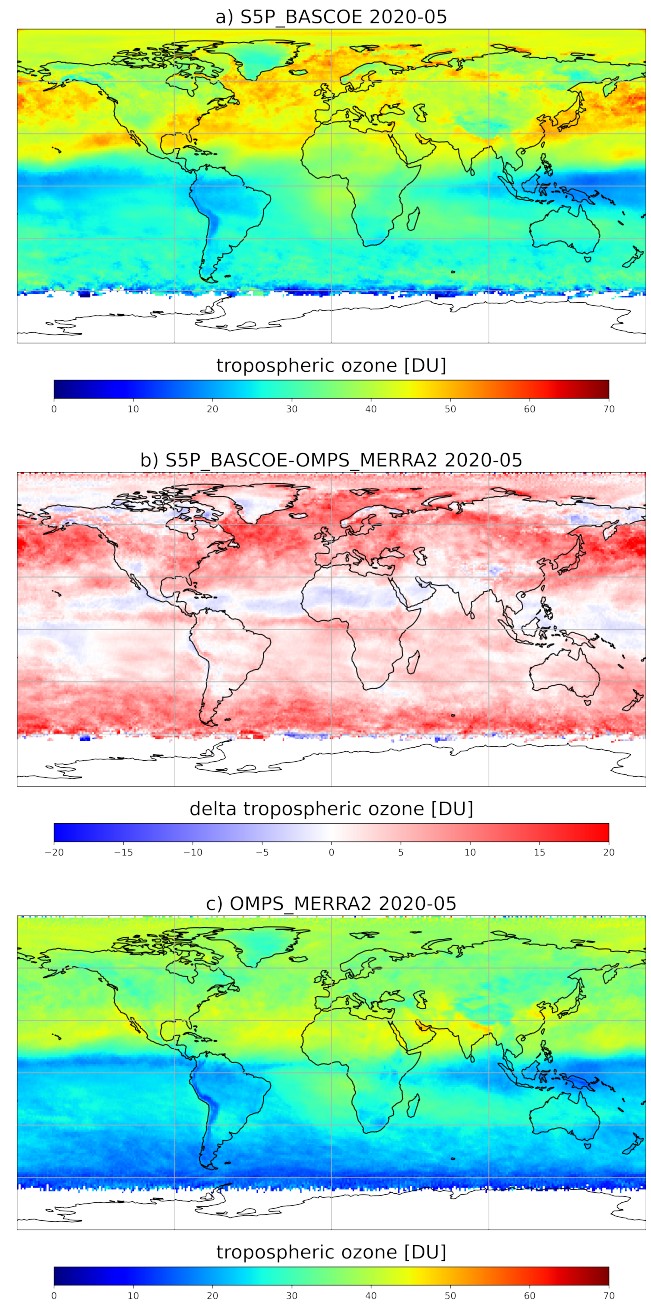

**Figure 6.** Monthly mean tropospheric ozone columns for May 2020 as observed by S5P-BASCOE (a) OMPS-MERRA-2 (c), and the difference between these two data sets (b).

of: 3.34 ± 7.64 DU. Garane et al. (2019) showed that the TROPOMI NRTI total ozone column is overestimated by ≈1 % or 4 DU relative to Brewer and Dobson spectrometers. Compared to OMPS we can find a similar deviation in the total columns

(not shown). Therefore the deviation in the tropospheric ozone column in a similar order of magnitude is to be expected. The time series of both tropospheric ozone products S5P-BASCOE and OMPS-MERRA-2 (not shown), reveal that the three peaks in June and July 2018 are partly caused by a decrease in the OMPS-MERRA-2 data set and to some extent by an increase in the S5P-BASCOE data. The different stratospheric ozone models BASCOE-FD (Sec. 2.3) and MERRA-2 (Sec. 3.1.2) are both constraint by MLS ozone profile observations. Nevertheless, some differences can be found. For BASCOE a small ozone deficit is known and corrected for. The correction in Figure 2 ranges between -1 and 3 DU, in the mean it contributes 1 DU to the stratospheric column. For the tropospheric columns this causes a corresponding reduction.

The monthly mean difference between S5P-BASCOE and OMPS-MERRA-2 as shown in the centre of Figure 6 shows a very good agreement between ≈30°N and 30°S. Also figure 5 confirms a better agreement in the tropics. Higher deviations occur over the northern Atlantic and Pacific Oceans up to 15 DU. Relative to the surrounding areas the S5P-BASCOE columns increase in the regions of the maximum difference and the OMPS-MERRA-2 data decrease in these regions (e.g. between Ireland and the central Atlantic Ocean). The size of the structures and the local changes depending on the data set indicates that it might be related to cloud data. While for S5P-BASCOE the total columns are cloud filtered and only data with cloud fractions less than 20 % are used, no cloud filter is applied during the OMPS-MERRA-2 retrieval.

In BASCOE the tropopause level is given as the lowest layer with a PV value higher than 3.5 PVU, whereas in MERRA-2 the 2.5 PVU level is used. This means that the tropopause in the S5P-BASCOE data set is higher compared to the OMPS-MERRA-2, and hence the tropospheric column is expected to be higher. Within the tropics, where differences between the two tropospheric column data sets are smallest, both BASCOE and MERRA-2 use the 380 K potential temperature definition. For the BASCOE data after 1 August 2019 also the 2.5 PVU pressure is given in addition to standard definition. This can be used to calculate the difference cause by the different tropopause definitions, the monthly mean May 2020 is shown in Figure 7. In the tropics the 380 K level is used, however a small difference in the definitions might cause the differences in both pressure and tropospheric ozone. In BASCOE the tropopause level is given as the centre of the pressure level containing the 3.5 PVU or 380 K level, for the comparison the 2.5 PVU/ 380 K pressure level is given directly. The global mean difference between these two tropospheric ozone data equals 1.82 DU and might therefore explain the differences between our tropospheric ozone data set and OPMS-MERRA-2 to a large extend. Moreover, the general pattern of the difference agrees well with the pattern observed in Figure 6. Both figures show a negative deviation in the tropics and positive one in mid-latitude. The remaining differences (figure 7 c)) between S5P-BASCOE(2.5 PVU) and OMPS-MERRA-2 are caused by either differences in the total columns (1-2 DU are below 1 % of the total column) or by other differences in the stratospheric ozone columns.

### 3.2.3 Comparison to Sondes

We compare the ozone sonde data for the period April 2018 to October 2020 with collocated satellite observations. We assume data to be collocated if the sounding was on the same day and the distance between the sounding station and the satellite observation was less than 25 km. The sonde data are integrated from the ground level to tropopause. The tropopause pressure (3.5 PVU) is read from the collocated S5P-BASCOE files and the mean tropopause pressure is used as upper limit for the sonde integration. We have to be aware that the surrounding may be heterogeneous with respect to urban and rural areas or

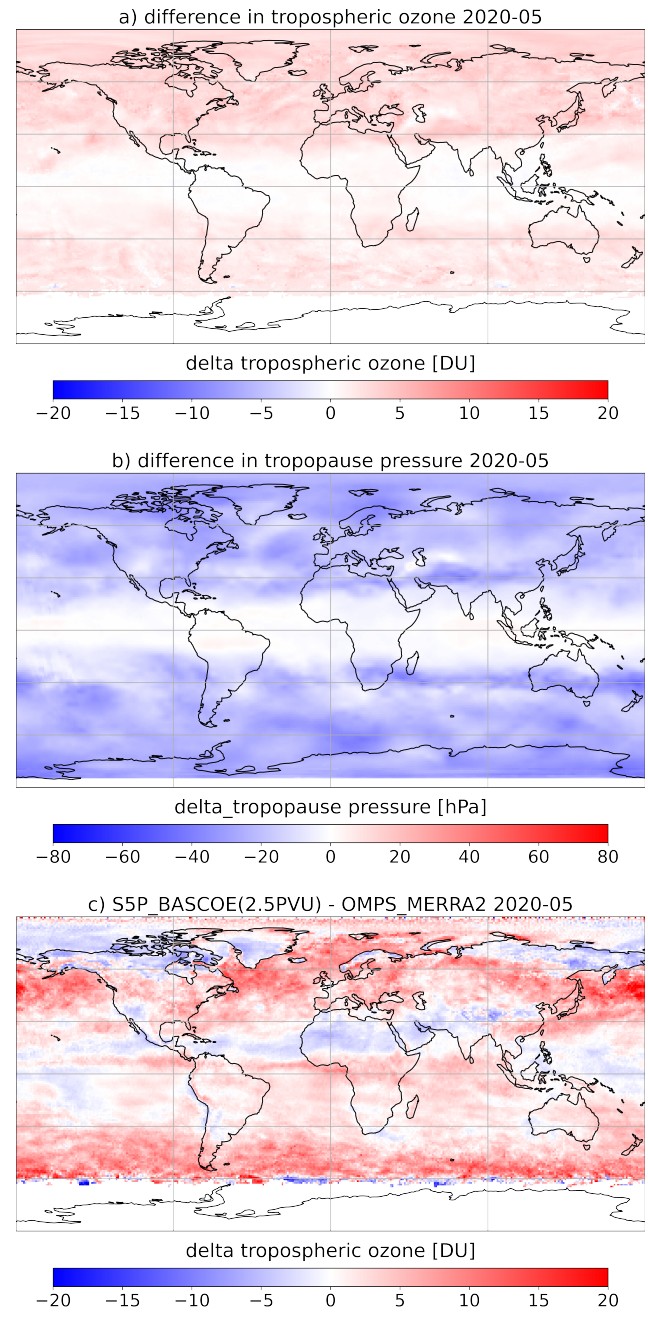

**Figure 7.** Difference in the BASCOE tropopause pressure $P_{tropopause}$(3.5 PVU)-$P_{tropopause}$(2.5 PVU) (b) and the respective tropospheric ozone columns (a), for May 2020. Sub-figure (c) shows the difference to OMPS-MERRA-2 (compare figure 6 b)) when using the 2.5 PVU tropoause.

mountains (Figure 8) or sea. To reduce this effect we used a 25 km radius, for comparison a 100 km is indicated in Figure 8 as well.

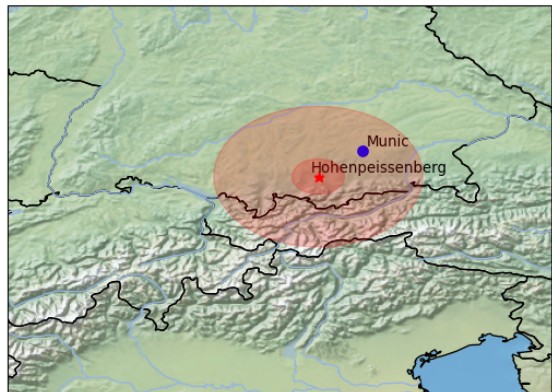

**Figure 8.** Southern Germany and the Alps including the location of the Hohenpeißenberg sounding station and a circle of 25 km to illustrate the size of the area sampled by the satellite. For comparison a second circle with 100 km radius is shown. The larger circle includes both urban (Munich) and alpine regions. Due to the projection the circles are slightly distorted. Image made using Natural Earth.

Sometimes Brewer or Dobson instruments are situated next to the sounding station and the respective total column data are provided together with the sonde profile. This allows us to compare both total and tropospheric ozone column. Thereby a potential deviation of the total column that might affect the tropospheric column can be detected.

For the sonde validation at Hohenpeißenberg shown in Figure 9 an overestimation in the winter / spring season is observed. A deviation in the total columns due to the enhanced albedo in winter is already documented by Inness et al. (2019) and Garane et al. (2019). An algorithm update including a surface albedo retrieval (Loyola et al., 2020) improved the total columns significantly. However a small positive bias is still observed between the TROPOMI total column and the sondes. This deviation propagates into the tropospheric column. On the other hand there might also be an underestimation in the sonde data as W. Steinbrecht (DWD-Hohenpeißenberg) pointed out during the CEOS Atmospheric Composition Virtual Constellation Conference in June 2021. At some sonde stations the data providers integrate the data up to the top of atmosphere, assuming a climatology above the burst altitude, and compare it with nearby total column observations e.g. from Dobson spectrometers. The measured mixing ratios are scaled according to the ratio of the total columns. This scaling is quite common though not used in general (e.g. Logan et al., 2012). It helps harmonizing the data for long term time series it also corrects for short term variations and artificial drifts. The scaling factors vary between 0.8 and 1.2. However, if the ozone effective temperature is not considered in the Dobson spectrometer data retrieval, the retrieval might result in slightly smaller total ozone column, especially in the winter month. In this case also the scaled sonde data is underestimated.

The mean deviation per 10° latitude band (Figure 10) also shows a small positive bias, both in the tropics and the mid latitudes. The deviation increases towards the poles, especially for the comparison between 70° and 80° North. Similarly for the southern polar region a systematic bias in the total ozone column was found, which was partly caused by the above

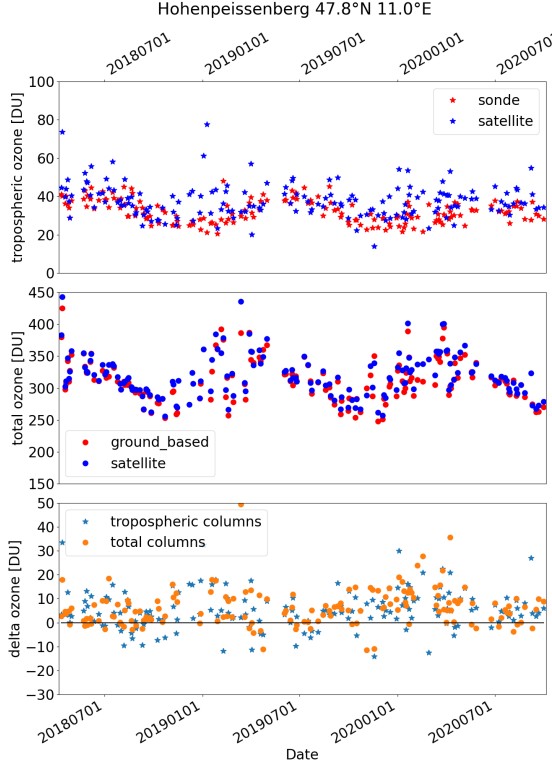

**Figure 9.** Comparison of S5P-BASCOE tropospheric ozone columns with ozone sondes at Hohenpeißenberg. On top the tropospheric columns are compared to the integrated sonde measurements. In the centre the total ozone columns are compared, finally at the bottom the differences between the satellite data and the integrated sondes data is shown.

mentioned albedo uncertainty. However, the bias is smaller here and for the tropospheric ozone columns it seems even less. The comparisons in the polar regions have to be taken with care, due to the sparse sampling in time and space, the comparison is certainly not representative. A small positive bias ($\leq 5$ DU) relative to the sonde data is found for tropical to mid-latitudes, details are shown in table 2. The global mean deviation equals 2.59 DU or 12.81 % with a standard deviation of 9.32 DU or 29.11 % for the 2254 comparisons used in this study. Based on the comparison we suggest using our data mainly between 50° S and 60° N.

## 4 Results

In the prevoius sections we introduced a new TROPOMI/S5P tropospheric ozone product and compared it to similar satellite products as well as to integrated ozone sondes measurements. In the following we will discuss the tropospheric ozone columns for some specific regions.

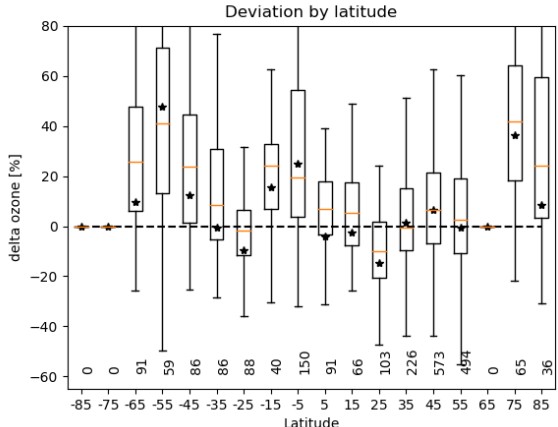

**Figure 10.** Box-Whisker plot showing median deviations (red line) between S5P-BASCOE and the sonde data for $10°$ latitude bands and the time period between April 2018 and October 2020. The boxes represent 25 and 75 percentile, the stars indicate the mean deviation of the tropospheric observations closest to the stations. The number at the bottom indicate the amount of comparisons per latitude band.

## 4.1 Global Tropospheric Ozone Distribution

Figure 11 shows the global tropospheric mean ozone distribution for the four seasons. All data from March, April and May
and the years 2018 to 2020 were average for the first subplot, and respectively for the other subplots. During the northern hemispheric spring the tropospheric ozone column is enhanced over the northern oceans. During the northern hemispheric summer three major enhancements can be seen: South Eastern US (section, 4.4), Eastern Mediterranean (section 4.3) and North Eastern China (not discussed here). In the tropics the typical wave one-pattern is found through-out the year, showing the global minimum in the Pacific Ocean north of New Guinea and the maximum is in the central Atlantic Ocean close to the
central African coast, however the amplitude varies with the season and is strongest in September to November. In the southern mid latitudes no significant structure is found, only a slight general increase in southern hemispheric spring.

Whether the enhanced ozone columns over the Northern Pacific Ocean between China, Japan and Alaska as well as over the Atlantic Ocean East of the US are caused by transport or other phenomena has to be investigated in future studies. Also cloud coverage and height influence the observed tropospheric ozone pattern, as already discussed in section 3.2.2. West of
355 the Iberian peninsula over the Atlantic Ocean and West of California over the Pacific Ocean smaller transport plumes are found though out the year with varying amplitude and latitude. The Californian one reaches the maximum in spring while the European / north African one in summer

## 4.2 Africa and tropical Atlantic

Biomass burning emits both VOCs and NOx which are the main precursors of tropospheric ozone. Africa contributes about
360 half of the total biomass burning carbon emissions (e.g. Pan et al., 2020). The local burning seasons moves north and south

**Table 2.** Mean difference and standard deviation per 10° latitude band, for the same data as displayed in Figure 10, where the median deviation is shown.

| Latitude band | number of comparisons | mean within 25 km radius | | standard deviation | |
|---|---|---|---|---|---|
| ° North | | % | DU | % | DU |
| 90- 80 | 36 | 32.87 | 8.38 | 42.21 | 10.17 |
| 80-70 | 65 | 48.44 | 12.07 | 43.60 | 8.75 |
| 70-60 | 0 | – | – | – | – |
| 60-50 | 494 | 5.83 | 0.35 | 27.36 | 13.43 |
| 50-40 | 573 | 10.22 | 2.93 | 26.46 | 8.35 |
| 40-30 | 226 | 6.34 | 1.25 | 30.98 | 9.18 |
| 30-20 | 103 | -7.69 | -4.84 | 21.59 | 7.99 |
| 20-10 | 66 | 6.19 | 1.44 | 19.13 | 5.44 |
| 10-0 | 91 | 7.75 | 1.71 | 19.25 | 4.67 |
| 0 - -10 | 150 | 28.31 | 7.28 | 29.33 | 7.45 |
| -10 - -20 | 40 | 21.12 | 4.08 | 23.98 | 4.83 |
| -20 - -30 | 88 | -1.11 | -1.06 | 17.93 | 5.84 |
| -30 - -40 | 86 | 13.42 | 2.98 | 31.24 | 7.72 |
| -40 - -50 | 86 | 27.45 | 5.53 | 38.17 | 7.38 |
| -50 - -60 | 59 | 43.21 | 7.91 | 41.75 | 7.27 |
| -60 - -70 | 91 | 32.87 | 5.65 | 38.75 | 6.41 |
| -70 - -90 | 0 | – | – | – | – |

following the ITCZ, with a time shift of 6 months. The tropospheric ozone columns over the tropical Atlantic reach a maximum in Sep-Nov season (figure 11). Figure 12 shows the VIIRS fire counts (https://firms.modaps.eosdis.nasa.gov/download/, May 2022) and the respective S5p-BASCOE tropospheric ozone distribution. It is remarkable that the ozone column over the African Continent is lower compared to the Atlantic ocean. The low sensitivity of TROPOMI to ozone in the lower troposphere might cause an underestimation if the ozone concentration is enhanced close to ground. Tropospheric ozone over the tropical Atlantic is caused by combination of lighting $NO_x$ emissions and biomass burning emission in both Africa and South America combined with uplift and long range transport. According to Moxim and Levy (2000), the polluted air masses rise over the continents and they are transported over the ocean where they subside. During the transport $NO_x$ from lightning and biomass burning react with VOCs to ozone. Sofieva et al. (2022) included chemical transport models in their study and confirmed the enhanced columns over the Southern Atlantic in the middle troposphere. They also found low tropospheric columns over the African continent that can be attributed to the low sensitivity of UV nadir viewing satellites for boundary layer trace gases.

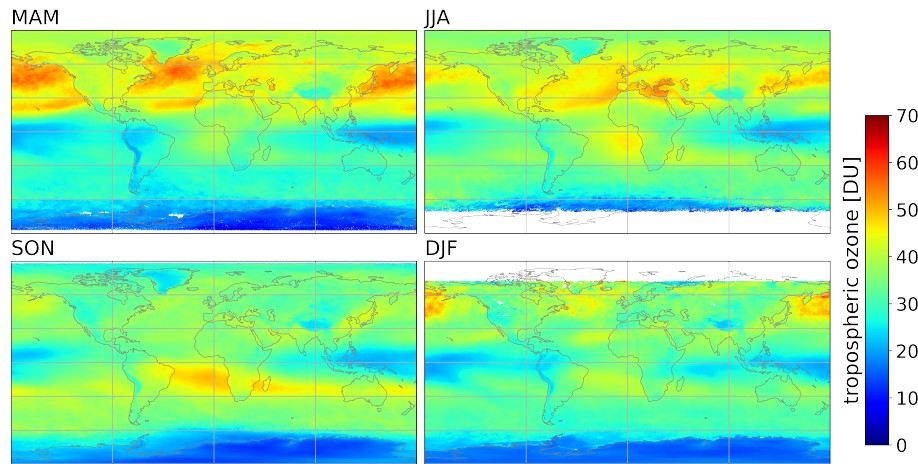

**Figure 11.** Global distribution of the mean tropospheric ozone for the four seasons obtained from S5P-BASCOE between 2018 and 2020.

### 4.3 Europe and the Mediterranean

The people living around the Eastern Mediterranean regularly suffer from high ozone concentration in summer (e.g. Dayan et al., 2017).

The time series of tropospheric ozone columns over the Greece capital of Athens (Figure 13) shows enhanced values for the summer 2018 and 2019 reaching up to 80 DU. In July and August several days of high column density are observed, the lowest values are still well above 40 to 50 DU. Enhanced column density are also found through-out the years, but especially in spring the column density often decreases rapidly after a few days. Such a decrease is hardly observed in summer. Similar time series can be found at several places around the Eastern Mediterranean, indicating stable conditions for a longer period.

According to Figure 11 the high ozone values reach from the Eastern Mediterranean to the Persian Gulf. The summer 2019 was extremely dry in northern Germany and large parts of Europe and the weather was stable for a week or two (https: //www.dwd.de/DE/wetter/thema_des_tages/2019/12/21.html, Jan. 2022). However, the time series for Berlin (Figure 13) still shows lower tropospheric columns and a smaller amplitude in the seasonal cycle but the day to day variation in the tropospheric ozone column seems higher in Berlin compared to Athens.

### 4.4 Southern United States

In the South West of the United States high ozone columns are observed in summer. The observed tropospheric ozone columns over the United States are shown in Figure 14. High tropospheric ozone columns are found east of 100° West, this correlates very well with the enhanced formaldehyde (HCHO) columns as observed by S5P (de Smedt et al., 2018). Formaldehyde can be used as tracers for VOCs as tropospheric ozone precursors. The maximum in the tropospheric ozone is shifted to the east

compared to formaldehyde. Due to longer lifetime of ozone the tropospheric ozone is transported to the north east and over the Atlantic Ocean. According to sonde and airborne observations (e.g. Cooper et al., 2007) such enhancements are observed

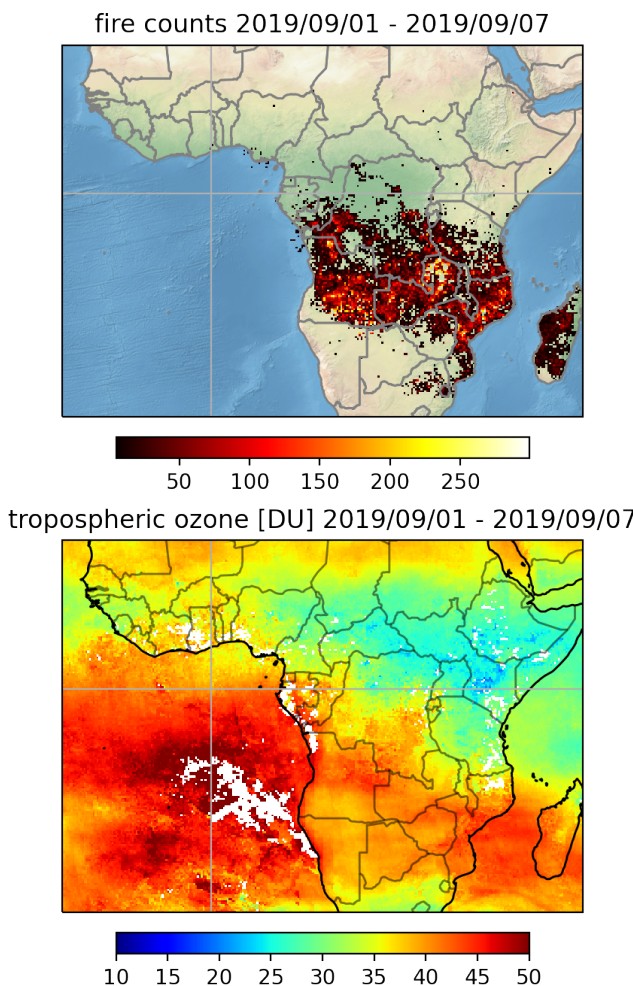

**Figure 12.** Active VIIRS fire counts for 1-7 September 2019 over Central Africa (top) and mean S5P-BASCOE tropospheric ozone columns same period (bottom). Fire data can be downloaded at https://firms.modaps.eosdis.nasa.gov/download/, last access May 2022

regularly in this region and are to large extend caused by the uplift of VOC rich air and the mixing in of lighting $NO_x$. A similar spatial pattern is observed over California; here the maximum of the tropospheric column is clearly separated from the HCHO enhancements.

## 5 Conclusions

We presented a new tropospheric ozone dataset based on Sentinel 5P/TROPOMI total ozone columns in combination with BASCOE stratospheric columns. The S5P-BASCOE tropospheric columns have the high TROPOMI spatial resolution of up to 3.5 x 5.5 km$^2$ and are in good agreement (3.34 $\pm$ 7.64 DU) with the tropospheric ozone data based on OMPS/MERRA-2.

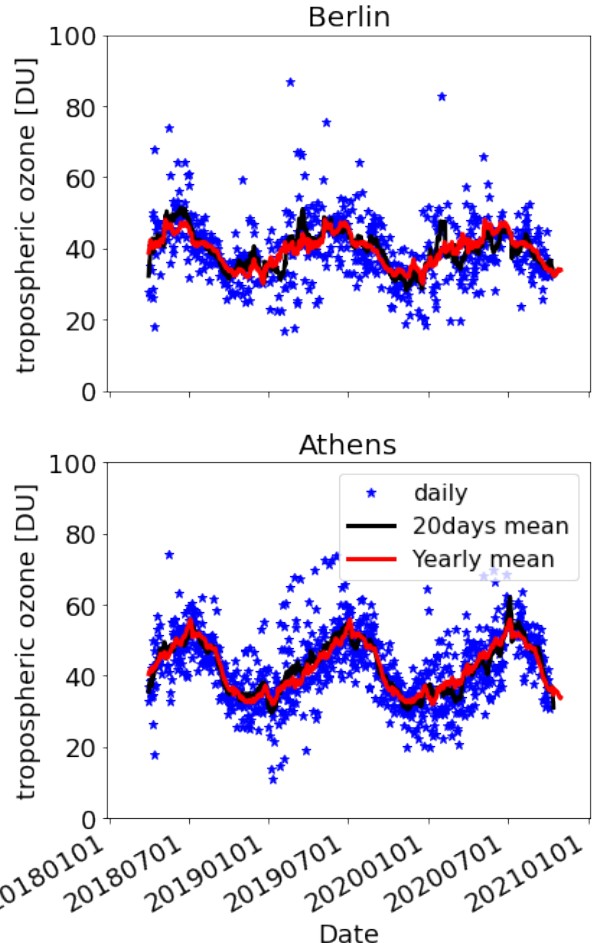

**Figure 13.** S5P-BASCOE tropospheric ozone column over Athens (Greece) and Berlin (Germany) (50 km radius around the city centre). The time series shows clear maxima in summer and minima in winter as expected. The blue stars indicate daily observations, while the black line is the 20 days running mean. For the comparison between the different years the typical annual cycle in red is included based on the 20 days running mean of each year.

Differences in the total column and different tropopause altitude cause the observed difference. S5P_CCD and S5P-BASCOE
cover a different altitude range. For the comparison we added a correction column based on a climatology (McPeters and Labow , 2012) to the CCD data. In the mean S5P-BASCOE and the S5P_CCD data agree very well with a deviation of -0.91 $\pm$ 5.76 DU. The comparison shows a larger standard deviation as to OMPS-MERRA2, and at least for the current time range an annual cycle is seen. Comparison of S5P_CCD with other datasets suggest that, this might be driven by the annual cycle in the S5P_CCD dataset. The algorithms of OMPS-MERRA-2 and S5P-BASCOE are very similar, while the S5P_CCD data
are retrieved using a completely different approach. The comparison to the ozone sondes showed a slight positive bias (2.6 $\pm$ 9.3 DU). Which might partly be caused by a small overestimation in the total column data, partly to an underestimation

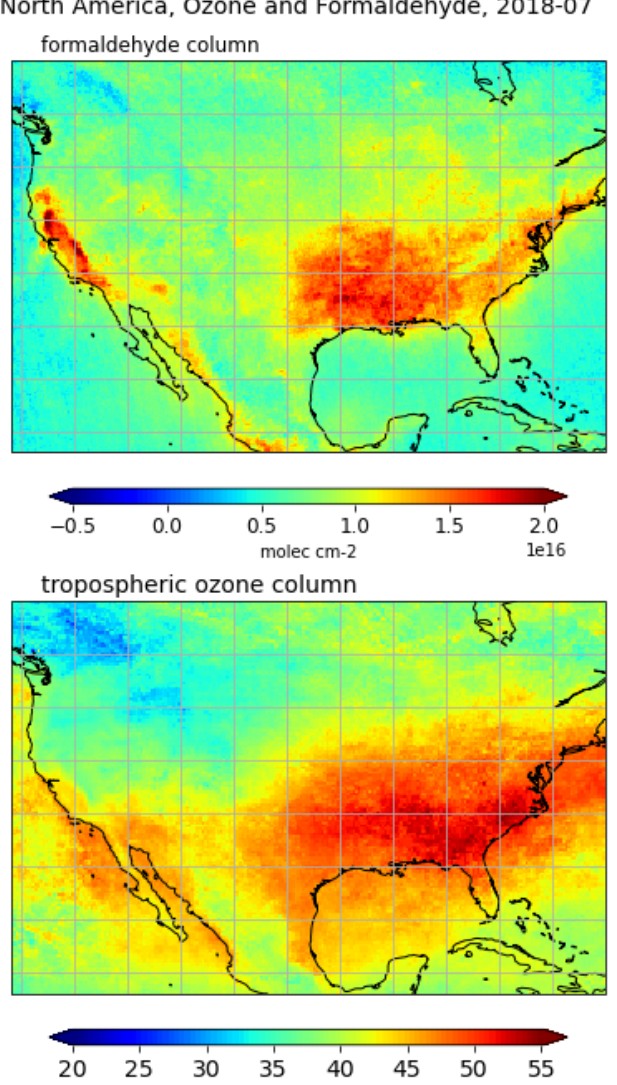

**Figure 14.** Monthly mean value of S5P formaldehyde (top) and S5P-BASCOE tropospheric ozone (bottom) over the United States observed in July 2018. Formaldehyde is a tropospheric ozone precursors.

from the sondes, the main reason however is the tropospheric column itself. Relative to the sonde data the difference increases towards the polar regions.

The S5P-BASCOE tropospheric ozone columns showed the expected global distribution. In the tropics the wave one pattern is found. During the northern hemispheric summer, the tropospheric ozone increases over the eastern Mediterranean or the South East of the United States. Some ozone enhancements over the Atlantic Ocean are attributed to medium range transport.

During the COVID lock down measures the emissions of tropospheric ozone precursors like $NO_x$ declined (e.g. Elshorbany et al., 2021). Due to the non linear $NO_x$-VOCs-ozone chemistry the $NO_x$ reduction does not necessarily lead to a reduction in tropospheric ozone. However, natural variability of the tropospheric ozone columns due to changes in the meteorological conditions is high. Here a longer time series of tropospheric ozone is essential to estimate natural variability, which might be in the order of the change caused by the COVID lock down. Because of that we plan to apply the same algorithm to past and current nadir satellite ozone observations from GOME-2 and OMI. Recent studies (Thompson et al., 2021; Ziemke et al., 2019) report an increase in tropospheric ozone, with a harmonised long term time series these findings can be verified.

*Data availability.* Currently the S5P-BASCOE data are available on request, we plan setting up a mapping and dissemination infrastructure.

*Author contributions.* This work is only possible within a team. Klaus-Peter Heue developed the S5P tropospheric ozone algorithms presented here and prepare this paper. Diego Loyola initiated this study and supported it with numerous discussions; he contributed to this paper through various helpful comments. Walter Zimmer and Fabian Romahn are responsible for operational UPAS implementation of the Sentinel 5P total ozone retrieval, the CCD retrieval and the S5P cloud retrieval. Simon Chabrillat and Quentin Errera provide the BASCOE ozone profile data and prepared the respective section in the manuscript. Jerry Ziemke and Natalya Kramarova provided the OMPS-MERRA-2 tropospheric data, including the respective section.

*Competing interests.* The authors have the following competing interests: At least one of the (co-)authors is a member of the editorial board of Atmospheric Measurement Techniques.

*Financial support.* This research was funded by the German Aerospace Center (DLR) in coordination with the DLR Innovative Products for Analyses of Atmospheric Composition (INPULS) project.

*Acknowledgements.* Thanks to Yves Christophe (retired from BIRA-IASB) for setting up the operational process of MLS assimilation by the BASCOE system.

We gratefully acknowledge all ozone sonde data providers for providing the $O_3$ profiles used in this manuscript: WMO/GAW Ozone Monitoring Community, World Meteorological Organization-Global Atmosphere Watch Program (WMO-GAW)/World Ozone and Ultraviolet Radiation Data Centre (WOUDC) https://woudc.org, last access Dec 2021. A list of all contributors is available on the website: https://doi.org/10.14287/10000001 We thank NOAA for providing the ozone sonde data from Boulder, Huntsville and Trinidad Head (ftp://aftp.cmdl.noaa.gov/data/ozwv/Ozonesonde, March 2022).

Thanks to EU/ESA/DLR for providing the operational TROPOMI/S5P products used in this paper: total ozone, CCD tropospheric ozone and cloud properties.

We acknowledge the use of data from NASA's Fire Information for Resource Management System (FIRMS) part of NASA's Earth Observing System Data and Information System (EOSDIS), https://earthdata.nasa.gov/firms, last access May 2022.

The comments and suggestions by the reviewers and by O. Cooper are gratefully acknowledged.

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
