# Peer review of "Tropospheric ozone retrieval by a combination of TROPOMI/S5P measurements with BASCOE assimilated data"

_Atmospheric Measurement Techniques, 2022_

## Referee Comment (RC1)

Review of the paper "Tropospheric ozone retrieval by a combination of TROPOMI/S5P measurements with BASCOE assimilated data" by Klaus-Peter Heue et al.

The manuscript presents a new way to retrieve global tropospheric ozone by mainly using TROPOMI NRTI total ozone and BASCOE assimilated stratospheric ozone profiles. A bias correction climatology was used to reduce BASCOE deficits in the upper stratosphere.
The retrieved ozone column is mapped on the TROPOMI measurement spatial resolution by interpolating BASCOE sub columns in space and time to TROPOMI. Results are validated with ozone sondes, TROPOMI CCD tropical ozone columns, and OMPS-MERRA2 data. A short section discusses the results for different regions of the Earth. I recommend publication after addressing the comments given below.

Does the paper address relevant scientific questions within the scope of AMT?
Yes

Does the paper present novel concepts, ideas, tools, or data?
Yes

Are substantial conclusions reached?
Yes

Are the scientific methods and assumptions valid and clearly outlined?
This needs to be improved. See comments below.

Are the results sufficient to support the interpretations and conclusions?
Yes

Is the description of experiments and calculations sufficiently complete and precise to allow their reproduction by fellow scientists (traceability of results)?
No formula was given in the paper.

Do the authors give proper credit to related work and clearly indicate their own new/original contribution?
Yes

Does the title clearly reflect the contents of the paper?
Yes

Does the abstract provide a concise and complete summary?
Yes but it should be upgraded a bit: Some typos (TROMOMI,..) and missing information (tropopause level?)

Is the overall presentation well structured and clear?

The presentation is OK but can be improved (see comments).

Is the language fluent and precise?
Quite some errors throughout the paper that need to be taken care of.

Are mathematical formulae, symbols, abbreviations, and units correctly defined and used?
No formulae, symbols/units must be streamlined (see comments)

Should any parts of the paper (text, formulae, figures, tables) be clarified, reduced, combined, or eliminated?
The subsection in section 5 are rather short, could be compressed into one? Otherwise see comments below.
A table/overview of bias/dispersion of BF vs OS, CCD, OM2 might help to get a clear view on accuracy/precision.

Are the number and quality of references appropriate?
Yes

Is the amount and quality of supplementary material appropriate?
-

Questions/Comments:

Lines:
6: OMPS-MERRA-2 data (globally? time span?)
11: exceptional larger positive deviation (where?)
12: expected spatial and temporal pattern (what is expected?)
24: beginning of the mission in 2018 = wrong, 2018 op. phase E2
33: MERRA-2 = citation?
65: It makes use = the dataset? sounds strange
72: ozone sonde will be explained briefly = data?
77: total ozone column is retrieved = use of op. data? which one?
75: maybe reorder chapter 2.1 and 2.2
94: TROPOMI data were reprocessed internally at DLR = full mission reprocessing to 2.1?
101: to prepare the EU Copernicus Atmospheric Monitoring Service = for what?
102: reduced, as expected for the austral spring = few words of explanation can help
112: Is there a change in BASCOE bias over time or is the early version not used anymore?
118: climatology: explain (e.g. resolution in time/space)
Fig1: add resolution to caption (mixingratio in title)
120: Where does the PV/PT data comes from?
121: Outside the tropics = which latitude? Is there a sharp cut or a mixed region?
Fig.2: what is the bias/dispersion of MLS stratospheric columns? Can
    longitudinal variations be neglected?
127: BASCOE ozone deficit above 4 hPa = citation available?
131: 5.5 x 3.5 km2 = not in 2018! pixel size switch at 2019/08/06

132: observation time = UTC time changing for every pixel?

Fig3: BASCOE, right? add resolution to caption, corrected for bias?

where does the jump in column near date line (~170°) come from?

Why only show stratospheric column and not tropospheric in section 2.4?

142: Dec.2021: inspection date?

150: to correct what? what climatology was used and how?

151: Does the correction depend on difference between reference and cloud height?

152: Tropical clouds are not on average at 10km, especially convective clouds.
    The S5P cloud retrieval underestimates the height in comparison to e.g. CALIPSO.

155: what is the defined correction height? 10km or 280hPa.
    The operational S5P product is adjusted to 270hPa?

157: why certain bands?

168: climatology the same as used for cloud height correction?

170: who is responsible for the product?

177: what version 2.1 algorithm means?

191: Reference for sonde comparisons?

213: why the two step approach and not grid directly to the reference grid?

217: S5P ATBD says 270hPa?

Fig4: Latitudes missing.

220: looks like there is a land-sea contrast as positive bias is mostly over land

221: where is it documented?

224: negative bias and variability of about 4 DU. => bias = 4DU?

224: three peaks are found = and that means?

Fig5: checked how subset of OMPS-M2 in tropics compared?
        annual pattern vs CCD: are there explanations for it?
        Daily values? Global mean?

235: mean difference 3-4DU: why no exact value?

244: 3.5PVU means BASCOE will have higher columns, right?

246: which year

Fig6: figures too small, hard to read

Fig6: difference in North Atlantic can be up to 15DU.
     Large BASCOE trop. column mostly over oceans, any explanations?

Fig7: Date? difference 3.5-2.5?

252: that means the 3.5 PVU is too high and adds a lot of
    stratospheric ozone into the BASCOE tropospheric column?

258: which tropopause definition here? what does mean pressure mean?

264: where is the sonde station? I wouldn't call 25DU "slight".

270: Reference?

Fig8: which satellite sample a 100 km circle? S5P has a higher resolution.

271: how and why is the sonde data scaled?

273: how large are those biases? ~8DU is not that small.
    relative differences are used in S5P to describe biases.
    what is the overall bias/dispersion? is there a time dependency?

278: end of sentence is missing.

285: typical enhancements are e.g. Mediterranean: rephrase. why are they typical?

290: columns correlated with other parameter e.g. cloud height?
Fig12: same map borders would be good
302: how much can be attributed to the positive bias?
304: The Berlin 20d mean has a lower dispersion than Athens from my point of view.
Fig14: Monthly mean?
320: small bias (~2-4DU = 5-10%, max~15DU)? Total columns or tropopause definition?
    CCD similar bias ~4DU
323: what does variability in difference means?
325-327: doubling of sentences. slight bias ~5DU (fig10)?
- Conclusions should be carefully revised

GENERAL ISSUES
Errors: quite a few typos, misspellings. Text needs to be heavily redacted.
units = sometimes no space between number and unit
nouns = capital first letter e.g. Tropopause?
deg or ° = unify
reference = needs to be checked thoroughly (e.g. German verbs not shown correct, spacing at wrong places)

ERRORS (only subset, thorough inspection necessary):
5: extend these data record into the future = ?
8: ozone sonde data = sub columns?
20: IPCC, 2013(@)
72: next section = number?
77: The second step includes = rephrase, hard to understand
80: place = space?
88: (Clouds as Layers Loyola et al., 2018).
104: the changes * 3, rephrase
145: OFL -> OFFL
166: to the 380 K level 80 to 130 hPa for
225: 2* periods
228: Tropospheric ozone columns from OMPS-MERRA-2 are described = misleading, retrieval is described

---

## Author Response (AR1)

*We thank the reviewer for the comments and suggestion, and apologize for the misprints.*

Review of the paper "Tropospheric ozone retrieval by a combination of TROPOMI/S5P measurements with BASCOE assimilated data" by Klaus-Peter Heue et al.

The manuscript presents a new way to retrieve global tropospheric ozone by mainly using TROPOMI NRTI total ozone and BASCOE assimilated stratospheric ozone profiles. A bias correction climatology was used to reduce BASCOE deficits in the upper stratosphere.

The retrieved ozone column is mapped on the TROPOMI measurement spatial resolution by interpolating BASCOE sub columns in space and time to TROPOMI. Results are validated with ozone sondes, TROPOMI CCD tropical ozone columns, and OMPS-MERRA2 data. A short section discusses the results for different regions of the Earth. I recommend publication after addressing the comments given below.

Does the paper address relevant scientific questions within the scope of AMT?
Yes

Does the paper present novel concepts, ideas, tools, or data?
Yes

Are substantial conclusions reached?
Yes

Are the scientific methods and assumptions valid and clearly outlined?
This needs to be improved. See comments below.

Are the results sufficient to support the interpretations and conclusions?
Yes

Is the description of experiments and calculations sufficiently complete and precise to allow their reproduction by fellow scientists (traceability of results)?
No formula was given in the paper.

Do the authors give proper credit to related work and clearly indicate their own new/original contribution?
Yes

Does the title clearly reflect the contents of the paper?
Yes

Does the abstract provide a concise and complete summary?
Yes but it should be upgraded a bit: Some typos (TROMOMI,..) and missing information (tropopause level?)

Is the overall presentation well structured and clear?
The presentation is OK but can be improved (see comments).

Is the language fluent and precise?
Quite some errors throughout the paper that need to be taken care of.

Are mathematical formulae, symbols, abbreviations, and units correctly defined and used?
No formulae, symbols/units must be streamlined (see comments)

Should any parts of the paper (text, formulae, figures, tables) be clarified, reduced, combined, or eliminated?
The subsection in section 5 are rather short, could be compressed into one? Otherwise see comments below.
A table/overview of bias/dispersion of BF vs OS, CCD, OM2 might help to get a clear view on accuracy/precision.

Are the number and quality of references appropriate?
Yes

Is the amount and quality of supplementary material appropriate?
-

Questions/Comments:

Lines:

6: OMPS-MERRA-2 data (globally? time span?)
*globally for the time period where S5P-BASCOE is available -updated in the manuscript*

11: exceptional larger positive deviation (where?)
*mostly in the northern polar regions - updated in the manuscript*

12: expected spatial and temporal pattern (what is expected?)
*in the tropics a wave one structures is expected and in the mid latitudes an annual cycle with a summer maximum - updated in the manuscript*

24: beginning of the mission in 2018 = wrong, 2018 op. phase E2
*Updated*

33: MERRA-2 = citation?
*Gelaro et al., 2017 , updated*

65: It makes use = the dataset? sounds strange
*replaced by "The algorithm makes use of"*

72: ozone sonde will be explained briefly = data?
*updated*

77: total ozone column is retrieved = use of op. data? which one?
*NRTI operational data as described in section 2.2 data - updated in the manuscript*

75: maybe reorder chapter 2.1 and 2.2
*We fully understand the referee's suggestion, but want to keep the current order as we first give a brief overview, before going into details. This was emphasized by adding references to the respective subsections.*

94: TROPOMI data were reprocessed internally at DLR = full mission reprocessing to 2.1?
*Yes - that is the advantage when total ozone processor was developed at the same institute, however this data set is not public available*
*update to "the DLR internal full TROPOMI mission reprocessing"*

101: to prepare the EU Copernicus Atmospheric Monitoring Service = for what?
*replaced by :*
*This implementation of BASCOE was developed to improve the representation of stratospheric composition in the EU Copernicus Atmospheric Monitoring Service (CAMS), by providing independent analyses of ozone and five other species which are also observed by MLS (HCl, ClO, HNO3, N2O, H2O). These analyses are used to evaluate the analyses and forecasts of stratospheric ozone which are delivered operationally by CAMS (e.g. Ramonet et al., 2019) and also to verify research versions of the CAMS system where the stratospheric chemistry module from the BASCOE system is implemented into the CAMS system (Huijnen et al., 2016).*

102: reduced, as expected for the austral spring = few words of explanation can help
*The observed reduction in the stratospheric ozone is caused by the well known ozone hole. updated in the manuscript.*

112: Is there a change in BASCOE bias over time or is the early version not used anymore?

*The early version is not used anymore; the current version is used since 2016 and the bias reported here is stable. Here is the revised paragraph:*
*"An early version (q2.4) of BASCOE-FD has been evaluated against total ozone ground-based measurements, ozonesonde profiles and satellite profiles over the period 2009-2012 (Lefever et al., 2015). The agreement was usually within +/- 10 % but degraded to +/- 40 % in the tropical tropopause layer (TTL). The version used here (5.7) runs operationally since March 2016. It is evaluated every three months for the validation of the CAMS operational analyses, indicating stable biases which are usually smaller than 5 % in the middle stratosphere and 15 % in the TTL (e.g. Sudarchikova et al., 2021)."*

118: climatology: explain (e.g. resolution in time/space)
*The original resolution is 5° latitude and 1 day. the climatology was interpolated to 2.5° latitude and smoothed in time. The information was added to the text.*

Fig1: add resolution to caption (mixingratio in title)
*updated*
*we added the total and tropospheric column as suggested by the reviewer and combined the figures 1 and 3 together with these new ones in one overview plot.*

120: Where does the PV/PT data comes from?
*The PV and PT data are imported from ECMWF*

121: Outside the tropics = which latitude? Is there a sharp cut or a mixed region?
*The tropics extend to 30° N and S. In this region the maximum pressure level Max(P(PV=3.5), P(PT=380K)) is used. Outside the tropics only the P(PV=3.5) is used. There is no mixed region, we clarified it:*
*"The calculation of the tropopause pressure is done in two independent steps, first the PV, PT tropopause is calculated. Outside the tropics (outside 30° South to North) the tropopause is defined as the Potential Vorticity isosurface at 3.5 PVU and inside the tropics as the isentropic isosurface with a potential temperature of 380 K or 3.5 PVU, whatever is lower.*
*The second steps is based on the WMO (World Meteorological Organisation) definition, lowest level with dT/dz is <=2 K/km and remains <=2 K/km in the next 2 km. The potential vorticity and temperature are extracted from ECMWF operational analyses at a reduced spectral resolution (T31) corresponding to the coarse grid of BASCOE-FD.*
*In the final step the two definitions are combined by choosing the lower altitude / higher pressure level. For practical reasons the centre pressure level of the respective grid cell is given. "*

Fig.2: what is the bias/dispersion of MLS stratospheric columns? Can longitudinal variations be neglected?
*As stated by the MLS Quality and Description document (https://mls.jpl.nasa.gov/data/v4-2_data_quality_document.pdf , May 2022) the accuracy of MLS $O_3$ column values is around 4 % and the average biases between MLS and latitudinally-binned data from lidar and ozonesonde sites across the globe is typically within 5 % or better, with poorer behaviour at low latitudes in the UTLS. Added to the manuscript.*
*We have to admit we never looked into the longitudinal variations in the difference between BASCOE and MLS. Due to the fact that ozone concentration above 4 hPa is low we assume the longitudinal change to be of secondary interest.*

127: BASCOE ozone deficit above 4 hPa = citation available?

*Yes, this deficit is described in Errera et al., 2019, already cited in this context (l 117), more details are given in Skachko et al 2016. (referenced therein)*

131: 5.5 x 3.5 km2 = not in 2018! pixel size switch at 2019/08/06
*corrected*

132: observation time = UTC time changing for every pixel?
*The unit of time interpolation is not relevant here. Of course a consistent time unit has to be used between TROPOMI and BASCOE and other time-dependent data (e.g. external tropopause). The time of the S5P observation does not change for every pixel but for every scanline, however, during the interpolation the fact that all pixels in a scanline are take simultaneously is not used, thereby the algorithm can be easily transferred to other sensors e.g. GOME-2.*
*no change in the manuscript*

Fig3: BASCOE, right? add resolution to caption, corrected for bias?
*Fig 3 shows the stratospheric column (BASCOE including bias correction), interpolated to the S5P pixels, so it has the S5P resolution of 5.5 x 3.5 km. Clarified in the figure caption.*
*we added the total and tropospheric column and combined the figures 1 and 3 together with these new ones in one overview plot.*

where does the jump in column near date line (~170°) come from?
*The data shown are interpolated to S5P resolution in space and time. At 170°W you can see both the first and the last orbit of this day. The jump is caused by the time difference of ~23 hour between the measurements in this area. Clarified in the figure caption*

Why only show stratospheric column and not tropospheric in section 2.4?
*The referee is certainly right it would be interesting to show the tropospheric data as well. On the other hand, we already show many figures and examples for tropospheric columns are shown in section 5. Therefore we introduced the overview plot for the figure 1 and 3. together with total and tropospheric column.*

142: Dec.2021: inspection date?
*According to the AMT guidelines the access date has to given with the webpage, clarified.*

150: to correct what? what climatology was used and how?
151: Does the correction depend on difference between reference and cloud height?
*To normalize the above cloud ozone column for the varying cloud altitudes to the reference level, the partial ozone column between the cloud optical altitude and the reference level was added. This correction column is based on the climatology by McPeters and Labow (2012).*

152: Tropical clouds are not on average at 10km, especially convective clouds.
The S5P cloud retrieval underestimates the height in comparison to e.g. CALIPSO.
*The S5P cloud retrieval might underestimate the cloud top height, especially for ice clouds, but what is more important for the tropospheric ozone calculation is to use the same cloud properties as in the total column ozone retrieval. Therefore, we use the cloud data given in the total ozone data file.*

155: what is the defined correction height? 10km or 280hPa.
The operational S5P product is adjusted to 270hPa?
*This was typo in the manuscript, at the beginning of the S5P CCD project the reference level was at 280hPa but we changed it to 270hPa, somehow I kept the 280 here. Corrected*

157: why certain bands?
*corrected: "It is assumed that for each latitude band the stratospheric ozone column is constant along the longitude and varies only slowly in time." see also answer to review comments #2.*

168: climatology the same as used for cloud height correction?
*yes, added*

170: who is responsible for the product?
*corrected to S5P-BASCOE*

177: what version 2.1 algorithm means?
*The algorithm is similar to TOMS v8, more details are described by McPeters et al 2019. we added this information*

191: Reference for sonde comparisons?
*Elshorbany, Y. Y, H. C. Kapper, J. R. Ziemke, S. A. Parr, The Status of Air Quality in the United States during the COVID-19 Pandemic: A Remote Sensing Perspective, Rem. Sens., 13(3), 369, https://doi.org/10.3390/rs13030369. 2021. added*

213: why the two step approach and not grid directly to the reference grid?
*The S5P-BASCOE data are compared to two different products with different resolution and to reduce the effort of the gridding we used a smaller pixel size. Moreover, the gridded data where used for the monthly or weekly means shown in section 5. But the referee is right, 0.5° by 1° would be sufficient for our comparison approach*

217: S5P ATBD says 270hPa?
*corrected see comment l 155*

Fig4: Latitudes missing.
*updated*

220: looks like there is a land-sea contrast as positive bias is mostly over land
*There is indeed a larger positive bias over northern Africa but there is also a positive bias over the Pacific Ocean and negative bias over South America. If there was a land-sea contrast it would be most obvious in the long term mean. The figure below shows the mean deviation S5P-BASCOE - S5P_CCD for the complete comparison period (April 2018 to October 2020). We still see a positive bias over the Saharan dessert, but also over the Pacific Ocean. On the other hand, negative deviations show up over the Indian Ocean and the adjacent landmasses in Indonesia, but also over Africa.*

[Figure]

*Figure 1: Mean difference between S5P-Bascoe and the S5P_CCD data for the complete comparison period*

221: where is it documented?
*S5P Validation reports, added*

224: negative bias and variability of about 4 DU. => bias = 4DU?
*Clarified exact numbers for bias and deviation are given -0.91 ± 5.76 relative to CCD*

224: three peaks are found = and that means?
*Clarified,*
*The time series of the tropical averaged difference between S5P-BASCOE and S5P\_CCD is illustrated in Figure 6. The figure also includes a comparison to the OMPS-MERRA-2 tropospheric column which will be discussed in more detail in section 3.2.2. The daily averaged tropical differences to the CCD also show a negative bias -0.91 DU and a standard variation of 5.76 DU. The time series shows a high variability of about 10 DU peak to peak. Three smaller peaks are observed in June / July 2018 and large one in May / June 2019. Some of the peaks also occur in the comparison with OMPS-MERRA-2. The large peak in May / June 2019 results from a decrease in the S5P_CCD data in this period, the cause is not yet fully understood. Also for the first peak 10th June 2018 a deviation in the CCD data (not shown in the paper) contributes to the increase in the differences. For the next two peaks it seems to be an overestimation of the tropical ozone by S5P-BASCOE.*

[Figure]

*Figure 2: Absolute values of tropospheric ozone columns from CCD (orange) and S5P-BASCOE (blue for the tropics) for 2018-05-15 until 2012-08-01*

Fig5: checked how subset of OMPS-M2 in tropics compared?
*included, shows a few small peaks and no annual cycle*
annual pattern vs CCD: are there explanations for it?
*I am not sure this is really the case, in the absolute values the annual cycle agrees well between S5P-BASCOE, OMPS-MERRA2 and CCD, however the amplitude is in the range of 2 DU this might not be obvious in the absolute values. According to validation report of the S5P_CCD there are indications of an annual cycle in the deviation relative to some sounding stations and to other CCD data from OMI and GOME-2B.*
Daily values? Global mean?
*Yes, the global / tropical daily mean values are shown in the figure, clarified in the caption.*
*"Time series of the daily mean differences between S5P-BASCOE and S5P_CCD (green, tropical) and OMPS-MERRA-2 (blue, global or orange, tropical)"*

[Figure]

*Figure 3: Time series of the daily mean differences between S5P-BASCOE and S5P_CCD (green, tropical) and OMPS-MERRA-2 (blue, global or orange, tropical)*

235: mean difference 3-4DU: why no exact value?
*changed: 3.34 ± 7.64 DU for OMPS*

244: 3.5PVU means BASCOE will have higher columns, right?
*Yes, if the tropopause is set to 3.5 PVU instead of 2.5 PVU this means the tropopause is higher and hence the tropospheric column is higher, added to the manuscript*

246: which year
*August 2019 (see section 2.3) added*

Fig6: figures too small, hard to read
*Changed to a vertical arrangement - increase in size*

Fig6: difference in North Atlantic can be up to 15DU.
Large BASCOE trop. column mostly over oceans, any explanations?
*According to figure 8 (updated, tropopause pressure difference and related ozone difference for May 2020) the difference due to the tropopause pressure might reach up to 5 DU. The total column might also contribute a few DU (~2). This explains already half of the observed differences. Also cloud filtering might have an influence, especially as the cloud filter might favour high ozone columns over the Atlantic Ocean.*

Fig7: Date? difference 3.5-2.5?
*The difference 3.5 - 2.5 = 1*
*Do you mean whether the figure contains the difference in the $O_3$ subcolumns upto P(3.5 PVU) and P(2.5 PVU)?*
*Yes, it contains $P_{tropopause}$(3.5 PVU)-$P_{tropopause}$(2.5 PVU) and the corresponding ozone subcolumns*

$$\int_{P_{Surface}}^{P(3.5PVU)} O_3(p)\,dp - \int_{P_{Surface}}^{P(2.5PVU)} O_3(p)\,dp \quad respectively$$

*Following the recommendation of reviewer #2 the date was changed to monthly mean for May 2020, as in figure 6*

252: that means the 3.5 PVU is too high and adds a lot of stratospheric ozone into the BASCOE tropospheric column?
*No, it just means that different definitions of the tropopause, cause different tropospheric ozone columns.*

*On the other hand, the difference between S5P-BASCOE and OMPS-MERRA-2 can not be fully explained by the difference in the tropopause. Parts of the difference is also caused by difference in the total column (here 1-2 DU is less than 1% ) or by other difference in the stratospheric columns (beside the tropopause) - updated*

258: which tropopause definition here? what does mean pressure mean?
*(clarified) "The tropopause pressure (3.5 PVU) is read from the collocated S5P-BASCOE files and the mean tropopause pressure is used as upper limit for the sonde integration."*

264: where is the sonde station? I wouldn't call 25DU "slight".
*The sonde station is at Hohenpeißenberg as mentioned in the figure caption. clarified in the text. The word "slight" is removed. The respective sentence concerns total columns, where 25DU is about 6% which is within the acceptable range for outliers in the total columns.*

270: Reference?
*No, W. Steinbrecht explained it as chat-comment to my presentation.*

Fig8: which satellite sample a 100 km circle? S5P has a higher resolution.
*The reviewer is right the resolution of S5P is much higher. On the other hand parts of the area around the sounding stations might be hidden by clouds, and the data are affected by statistical noise. We averaged over a respective area to reduce the influence of noise and clouds around the sounding stations.*
*We study the influence of the radius in more detail, depending on the radius the number of comparisons is reduced, and the global mean deviation changes slightly. Based on this finding we reduced the radius to 25 km.*
*The table shows the dependency of the global mean deviation and the number of comparisons on the radius around the sounding stations.*

| radius | 100 km | 50 km | 25 km |
|---|---|---|---|
| number of comparisons | 3112 | 2687 | 2254 |
| mean difference [%] | 15.03 | 14.99 | 12.81 |
| standard deviation [%] | 30.18 | 30.26 | 29.11 |
| mean difference [DU] | 3.33 | 3.15 | 2.59 |
| standard deviation [DU] | 9.53 | 9.50 | 9.32 |

[Figure]

*Figure 4 same as figure 8 (old) in the manuscript with the different averaging radii mentioned in the table*

271: how and why is the sonde data scaled?

*The scaling of the sonde data is performed by the data providers.*

*First they integrate the sonde data to the total column and add a climatology based correction for the difference between burst altitude and top of atmosphere to get the total column. The total column is scaled to agree with spectrometer's observation. The same scale factor is applied to the mixing ratio at each altitude. According to Logan et al. (2012) the scaling is not applied to all sonde data in Europe and the scaling factor varied between 0.8 and 1.2.*

*Clarified:*

*"At some sonde stations the data providers integrate the data up to the top of atmosphere, assuming a climatology above the burst altitude, and compare it with nearby total column observations e.g. from Dobson spectrometers. The measured mixing ratios are scaled according to the ratio of the total columns. This scaling is quite common though not used in general (e.g. Logan et al., 12). It helps harmonizing the data for long term time series it also corrects for short term variations and artificial trends. The scaling factors vary between 0.8 and 1.2.*

*However, if the ozone effective temperature is not considered in the Dobson spectrometer data retrieval, this might result in a smaller ozone column, especially in the winter month. In this case also the scaled sonde data is underestimated."*

273: how large are those biases? ~8DU is not that small. relative differences are used in S5P to describe biases. what is the overall bias/dispersion? is there a time dependency?

*If the observable is close to zero the relative error, might be misleading, or not defined. Therefore for some trace gases e.g. $SO_2$ the absolute deviation might be a better choice. I am not sure if this is really the case for tropospheric ozone, here I think both the relative and the absolute deviations might be justified. With respect to the comment we partly changed to relative values for the sonde based comparison. Figure 10 (time series at Hohenpeißenberg) still uses absolute difference for both tropospheric and total ozone column, Figure 11 (box-whisker-plot) shows percental deviation. The table lists both, absolute and relative mean deviation per latitude.*

*Up to now we focused on roughly two years of data, this period is from my point of view not long enough to retrieve solid statistics about time dependencies, especially with respect to the sparse sampling for some of the sounding locations. For Hohenpeißenberg we see an overestimation in the winter period relative to the sondes.*

278: end of sentence is missing.

*Corrected:*

*"A small positive bias (~5 DU) relative to the sonde data is found for tropical to mid-latitudes, details are shown in table 1. The global mean deviation equals 2.8 DU or 13.5 % with a standard deviation of 9.4 DU or 29.9 % for the 2191 comparisons used in this study."*

285: typical enhancements are e.g. Mediterranean: rephrase. why are they typical?

*Typical in that sense, that they are observed regularly (each year) in the specific region and have been studied for many years now, seen comment by Owen Cooper for the enhancement over the US or Lelieveld et al. 2002 for the eastern Mediterranean.*

*clarified:*

*"Figure 12 shows the global tropospheric mean ozone distribution for the four seasons. All data from March, April and May and the years 2018 to 2020 were average for the first subplot, and respectively for the other subplots. During the northern hemispheric spring the tropospheric ozone column is enhanced over the northern oceans. During the northern hemispheric summer three major enhancements can be seen: South Eastern US (section, 4.4), Eastern Mediterranean (section 4.3) and North Eastern China (not discussed here)."*

290: columns correlated with other parameter e.g. cloud height?
*Besides transport other explanations like cloud structures might be possible as well, compare also to the difference to OMPS-M2 over the Atlantic Ocean.*
*Clarified*
*"Also cloud properties influence the observed tropospheric ozone pattern, as already discussed in section 3.2.2."*

Fig12: same map borders would be good
*updated, See also answer to referee #2*

302: how much can be attributed to the positive bias?
*The bias is not known everywhere and at any time. According to the update table (suggested by the reviewer) for this latitude band (30°-40°N) the mean bias is about 6.3% or 1.2 DU.*
*The extreme values of 80 DU are not the summer mean maximum, which is just above 60 DU. The respective bias can hence be estimated to less than 5 DU. We will leave this estimate to reader.*

304: The Berlin 20d mean has a lower dispersion than Athens from my point of view.
*clarified: the seasonal amplitude in Athens is larger as in Berlin, the short-term day to day variations however are higher in Berlin. See also comment to referee #2.*

Fig14: Monthly mean?
yes, the figure for the US shows monthly mean values. clarified

320: small bias (~2-4DU = 5-10%, max~15DU)? Total columns or tropopause definition?
*There are several different sources for deviations between the two data sets, the total column and the tropopause height are the most obvious ones. For S5P-BASCOE a cloud filter is applied, the data are gridded and interpolated between different resolutions for the final product as well as within the retrieval, this might also cause part of the observed structures.*
*Clarified in chapter "3.2.2 Comparison to OMPS-MERRA-2"*

CCD similar bias ~4DU
323: what does variability in difference means?
*this sentence refers to figure 5 (time series of S5P-BASCOE compared to CCD and OMPS-MERRA2). While the difference to OMPS-MERRA-2 changes only slightly in time the difference to the CCD data shows a much higher variability as a function of time.*

325-327: doubling of sentences. slight bias ~5DU (fig10)?
- Conclusions should be carefully revised
*updated*

GENERAL ISSUES
Errors: quite a few typos, misspellings. Text needs to be heavily redacted.
units = sometimes no space between number and unit
*corrected*
nouns = capital first letter e.g. Tropopause?
deg or ° = unify
*corrected*

reference = needs to be checked thoroughly (e.g. German verbs not shown correct, spacing at wrong places)
*corrected (French, German and Spanish special characters)*

ERRORS (only subset, thorough inspection necessary):
5: extend these data record into the future = ?
*deleted.*

8: ozone sonde data = sub columns?
*corrected, "integrated sonde data"*

20: IPCC, 2013(@)
*corrected*

72: next section = number?
*added*

77: The second step includes = rephrase, hard to understand
*changed: "In the second step the ozone profiles from BASCOE (section 2.3) are integrated between the tropopause pressure and the top of the atmosphere."*

80: place = space?
*changed*

88: (Clouds as Layers Loyola et al., 2018).
*removed "Clouds as Layers"*

104: the changes * 3, rephrase
*changed: "Since it is an operational service, the BASCOE-FD system has evolved with time due to the changes in the ECMWF operational system. Moreover BASCOE-FD was adapted to the updates in MLS retrieval algorithm and those in the BASCOE system (see the changelog here: http://www.copernicus-stratosphere.eu/4_NRT_products/3_Models_changelogs/BASCOE.php, Dec. 2021).*

145: OFL -> OFFL
*changed, also NRT to NRTI*

166: to the 380 K level 80 to 130 hPa for
*clarified*

225: 2* periods
*deleted one of them*

228: Tropospheric ozone columns from OMPS-MERRA-2 are described = misleading, retrieval is described
*clarified*

*Lelieveld, J., Berresheim, H., Borrmann, S., Crutzen, P. J., Dentener, F. J., Fischer, H., Feichter, J., Flatau, P. J., Heland, J., Holzinger, R., Korrmann, R., Lawrence, M. G., Levin, Z., Markowicz, K. M., Mihalopoulos, N., Minikin, A., Ramanathan, V., de Reus, M., Roelofs, G. J., Scheeren, H. A., Sciare, J., Schlager, H., Schultz, M., Siegmund, P., Steil, B., Stephanou,*

E. G., Stier, P., Traub, M., Warneke, C., Williams, J., and Ziereis, H.: Global air pollution crossroads over the Mediterranean, Science, 298, 794–799, 2002.

Logan, J. A., Staehelin, J., Megretskaia, I. A., Cammas, J.-P., Thouret, V, Claude, H., De Backer, H, Steinbacher, M., Scheel, H.-E., Stübi, R., Fröhlich, M., Derwent, R., Changes in ozone over Europe: Analysis of ozone measurements from sondes, regular aircraft (MOZAIC) and alpine surface sites, J. Geophys. Res., 117, D09301, doi:10.1029/2011JD016952, 2012.

Skachko, S., Ménard, R., Errera, Q., Christophe, Y., and Chabrillat, S.: EnKF and 4D-Var data assimilation with chemical transport model BASCOE (version 05.06), Geosci. Model Dev., 9, 2893–2908, https://doi.org/10.5194/gmd-9-2893-2016, 2016.  a, b, c, d, e, f, g, h

*We thank the reviewer for the comments and suggestion, and apologize for the misprints.*

GENERAL

The paper is dedicated to tropospheric ozone column retrieval using a combination of TROPOMI total ozone measurements with stratospheric ozone from the BASCOE assimilated data. The paper describes the retrieval algorithm, intercomparisons with other tropospheric ozone datasets, and the illustrations of geographical distributions.

The paper introduces an important work, and it contains important information. However, the presentation need significant improvements. Please find my comments below.

MAIN COMMENTS

1) The language must be improved: too many misprints and unclear formulations. Several of them (but not the full list) are in the "Detailed comment" and "Technical corrections".

2) The structure of the paper is not optimal, from my point of view.  It would be easier for reading, if comparison results would be placed immediately after a short description of other tropospheric datasets used for validation. I suggest: name Sect. 3: "Comparisons with other tropospheric datasets" with subsections like:

3.1. Comparison with  TROPOMI_CCD

3.1.1 TROPOMI_CCD dataset

3.1.2 Comparison results

3.2  Comparisons with OMPS-MERRA-2

 and  so on for other datasets.

*The suggested structure is reasonable as well.*
*Both these structures have their pros and cons and both can be found in the literature. The one suggested by the reviewer is sorted by the datasets but jumps between description and comparison. The one we used combines the comparisons, but the reader may have to go back to find some details on the dataset used for the current comparison.*
*After careful evaluation, we prefer to stick to selected structure: first explain the different datasets and then compare with S5P_BASCOE in a second step. But partially following the suggestion from the reviewer, we combine both sections in one .*

*3 Comparisons to other tropospheric ozone columns*
*3.1.other Observations*
*3.1.1 TROPOMI_CCD*
*3.1.2 OMPS-MERRA-2*
*3.1.3 Soundings*

*3.2 Comparison results*
*3.2.1 comparison to TROPOMI_CCD*

..

3) There is no information about S5P-BASCOE data availability.
*The data is not yet available to the user. This is planned in a next step that hopefully will be ready before the final version of the paper is published.*
*"Data availability. Currently the S5P-BASCOE data are available on request, we plan setting up a mapping and dissemination infrastructure."*

4) All acronyms should be explained at first appearance (note that the abstract is considered separately).
corrected

5) It would be advantageous showing more details of global tropospheric ozone morphology, in particular, global maps in different seasons. This would also support subsequent illustrations of tropospheric ozone in specific regions.
*added*

DETAILED COMMENTS

In Abstract, instead noting that the "algorithm is similar to the well established OMI-MLS or OMPS-MERRA-2 retrieval" (Lines 3-4), please say about the main principle of the retrieval (residual method). Information about temporal resolution and vertical coverage should be present in the abstract.
*corrected, added*
*"The BASCOE stratospheric data is interpolated to the S5P observations and subtracted from the TROPOMI total ozone column. Thereby a tropospheric residual ozone column from the surface up to the tropopause is gained. The tropospheric ozone columns are retrieved at the full spatial resolution of the TROPOMI sensor (5.5 x 3.5 km²) with a daily global coverage."*

Lines 31-32: A better formulation of the sentence is needed. MLS measures sometimes below the tropopause.
*The referee is certainly right MLS also observes the upper tropopshere. But we only use the stratospheric part of the profile here. We replaced "tropopause" by "upper troposphere". Thereby it is clear that also the upper troposphere is observed and in the following sentence it is stated that only the stratospheric part is used.*

Lines 33-35: For OMI-MLS, both original and with assimilation datasets are available, this is worth to state more clearly.
*clarified*

Lines 37-39: Is SCIAMACHY retrieval approach similar or different compared to OMI-MLS? A short note would be useful.
*yes the algorithm is similar, the limb observations are analysed and yield an ozone profile which is integrated later on, for the altitude range above the tropopause. added*

Lines 53-54: Please be consistent: for other datasets, the validation results are not mentioned.
*Validation results are removed*

Lines 58-62: A link to CAMS tropospheric ozone data would be useful.

*added: The CAMS ozone profiles can be downloaded at https://ads.atmosphere.copernicus.eu/cdsapp#!/search?type=dataset (last access March 2022).*

Lines 117-119: Is the bias stationary?
*No the bias varies within the yearly cycle and the latitude as obvious from figure 2*
*clarified*

Line 128: "the correction is in the order of 2 DU" Is this the same as in Figure 2?
*Yes, the correction added to the stratospheric column is illustrated in figure 2 (reference changed)*

Line 131: "TROPOMI/S5P has a daily global coverage with a spatial resolution of 5.5 x 3.5 km2". This information is given above in the text and its repetition is not needed here.
*I think it is useful to repeat this information here, because we have to combine the high resolution from total ozone columns and moderate resolution from BASCOE stratospheric columns. Hence we prefer to keep the repetition in this case.*

Lines 157-159. I believe that even within tropics this approximation (uniformity in longitude) has an uncertainty. If known, it would be good to specify it.

[Figure]

*Figure 1: BASCOE stratospheric ozone column for 2019-10-01 until 2019-10-06. Temporal mean 40°S to 40°N (right), longitudinal and temporal mean and standard deviation (left), the hashed area indicates the tropical band as used in the CCD.*
*The figure is not included in the manuscript.*

*This is the basic assumption of the CCD algorithm and the application of the CCD shows it's justification and limitations. The S5P_CCD algorithm uses the mean stratospheric ozone column for 6 days. InFigure 1 the mean BASCOE stratospheric column for 6 days is shown. For the range between 40°S and 40°N within the tropical band (20°S to 20°N) the longitudinal variation is small, also the error bars (indicating the standard deviation within 0.5 degree latitude and 6 days) are small compared to the higher latitudes.*
*added to the manuscript:*
*"It is assumed that for each latitude band the stratospheric ozone column is constant along the longitude and varies only slowly in time and latitude. This assumption is in general used for the CCD algorithm and is only justified within the tropics, therefore the algorithm is limited to the latitude range between 20° S and 20° N. For several examples of BASCOE stratospheric column varied by less than 5 DU standard deviation within 6 days, along the longitude for 0.5° latitude. The temporal and spatial resolution is comparable to the S5P_CDD settings."*

Line 196: OMPS-MLS -> OMPS-MERRA-2 ?
*corrected*

Lines 200-205:  Do I understand correctly that not all available ozonesonde data are used? A map showing location of sounding stations, preferably with colouring according to number of observations, would be useful.
*No, we used all soundings station available to us. Owen Cooper's comment included a link to two additional station in the United States i.e. Huntsville (Alabama) and Trinidad Head (California) a map (Figure 2) is included*

[Figure]

*Figure 2: Global distribution of ozone sounding station used in this study. The number behind the names indicate the number of sounding between April 2018 and June 2020, where S5P-BASCOE data were available within 25 km around the station.*

Lines 220-221: "The stripe in the south is caused by a well known and documented retrieval problem in the CCD data." A reference or an explanation would be useful.
*The stripe was removed by applying a better QA filter(0.7 instead of 0.5) in the comparison, a reference to the TROPOMI CCD PUM was added*

Figure 6 and discussion: Note also different spatial pattern, with large values over oceans in NH in S5P-BASCOE. Is this due to different tropopause height definition?  To evaluate this, difference in tropospheric ozone column in Figure 7 should be for the same period as in Figure 6, and presented with the same color scaling.
*corrected*

Line 246:  "For BASCOE data after August 1 "   which year?
*2019 included*

Lines 251-252 : "The influence of the different tropopause definitions on the tropospheric ozone is about 1-2 DU"   According to Fig.7,  it can be up to 10-15 DU.
*Yes, it can reach up to 15 DU but in the mean it is around 2 DU, clarified*

Lines 262-263: "This allows ...  and the potential deviations might be separated"  Rephrasing is needed
*changed to*
*"Sometimes Brewer or Dobson instruments are situated next to the sounding station and the respective total column data are provided together with the sonde profile. This allows us to compare both total and tropospheric ozone column. Thereby a potential deviation of the total column that might affect the tropospheric column can be detected."*

Lines 264-266:  "In version 2 of UPAS a new albedo retrieval scheme was implemented (Loyola et al., 2020) and respective comparison improved significantly." Are these improvements with respect to UPAS v 1?  If this is not show, there is no need to mention.
*changed. The improvements are indeed with respect to UPAS version 1, but this v1 was used in the validation paper by Garane et al. 2019, therefore it might be worth being mentioned here.*
*clarified:*
*"For the sonde validation at Hohenpeißenberg shown in Figure 10 an overestimation in the winter / spring season is observed. A deviation in the version 1 of the TROPOMI total ozone due to the enhanced albedo in winter was documented by Inness et al. (2019) and Garane et al. (2019). Version 2 of the TROPOMI total ozone includes a surface albedo retrieval (Loyola et al., 2020) that improved the total columns significantly. However, a small positive bias is still observed between the TROPOMI total column and the sondes. This deviation propagates into the tropospheric column."*

Line 268: "This deviation propagates into the tropospheric column" I do not see strong correlation between deviations of full and tropospheric ozone columns (Figure 9 bottom).
*The referee is right the correlation is not strong but the difference is systematically positive in the winter period, indicating an overestimation of satellite data or underestimation of the sondes in these periods.*

Line 270: Any reference on presentation by W. Steinbrecht?
*Unfortunately not, it was a chat comment by W. Steinbrecht to my presentation.*

Lines 271-272: "The ozone effective temperature is not considered in the Dobson spectrometer observations and the sonde data are scaled to the Dobson total ozone column."   What is the consequence for data quality?
*The sonde data are scaled by the data providers. (see also comment to reviewer #1)*
*The aim of the scaling is that the consistency between integrated sonde column and Dobson spectrometer is improved. It also reduces the day to day variability of the sonde measurements. However, if the effective temperature is not considered in the data analysis of the spectrometer, the total ozone column might be underestimated in winter. Because the same scaling factor is applied to the tropospheric sub-column, this might also be underestimated by sonde.*
*clarified:*
*"At some sonde stations the data providers integrate the data up to the top of atmosphere, assuming a climatology above the burst altitude, and compare it with nearby total column observations. The measured mixing ratios are scaled according to the ratio of the total columns. This scaling is quite common though not used in general [Logan et al., 12]. It helps harmonizing the data for long term time series it also corrects for short term variations and artificial drifts. The scaling factors vary between 0.8 and 1.2.*
*However, if the ozone effective temperature is not considered in the Dobson spectrometer data retrieval, the scaling might result in slightly smaller total ozone column, especially in the winter month. If the sondes underestimate the total column also the tropospheric column can be underestimated."*

Figure 10: The latitudes with zero collocations should be removed. Please add zero line and use better scaling. The caption says: "The stars indicate the mean of the tropospheric observations closest to the stations". Why some values indicated by stars are negative (for example, for 25N)? Is this correct? Deviations and absolute values should be shown either on different vertical axes with distinct colors, or stars should be removed.

*The stars do not indicate absolute values but the difference between the sondes and those satellite observations being closest to the station. Clarified and a zero line is added. The 10° latitudes bands with zero collocation however will remain, to avoid jumps in the x-axes and hence increase readability.*

Line 286: "In the tropics the typical wave one-pattern is found" Since this pattern is not related to wave activity, I believe, it should not be called "wave-one pattern".

*It is correct that this pattern is not caused by any wave activity, however the name is commonly used to describe the distribution of tropospheric ozone maximum and minimum in the tropics e.g. Ziemke 1998: "We also note that TCO amounts given in Table 1 corroborate the existence of the persistent tropical zonal wave 1 distribution [e.g., Fishman and Larsen, 1987;Ziemke et al., 1996; Hudson and Thompson, 1998] with high values in the Atlantic and low values in the Pacific." Therefore the fixed term will be kept.*

Sect 5.1. It would be advantageous to show also the seasonal dependence of total global maps. This would be useful in the discussions below. Figure 11 can have subplots corresponding to different seasons.
*added*

Sect. 5.2 Please explain the shift to the ocean and not observing strong ozone enhancement over Africa. *Following the reviewers suggestion we added four seasonal maps for the tropospheric ozone. These maps show that enhancement over the tropical Atlantic reaches the maximum in Sep-Nov. Because of that we replaced the tropospheric ozone and the fire maps by maps form the first week of September 2019 (Figure 3) instead of the last week of June 2019.*
*Also here the maximum in the tropospheric ozone column is situated over the Atlantic ocean and not above fire emissions. "The interaction of transport and chemistry" [Moxim and Levy, 2000] shifts the maximum ozone column out to the Atlantic.*
*Clarified: "Tropospheric ozone over the tropical Atlantic is caused by combination of lighting NOx emissions and biomass burning emission in both Africa and South America combined with uplift and long range transport. According to Moxim and Levy [2000], the polluted air masses rise over the continents and they are transported over the ocean where they subside. During the transport NOx from lightning and biomass burning react with VOCs to ozone."*

[Figure]

*Figure 3: Top: first week of September 2019, VIIRS fire data (https://earthdata.nasa.gov/earth-observation-data/near-real-time/firms/active-fire-data, May 2022) tropospheric ozone columns for the same period, bottom.*

Section 5.3: For Europe and Mediterranean, I suggest using the cities, for which also ground-based observations are available. Adding the curves from ground-based observations to Figure 13 would confirm the validity of S5P-BASCOE time series

*I checked the referee's suggestion to include ground based ozone measurements for Berlin and Athens (https://discomap.eea.europa.eu/map/fme/AirQualityExport.htm, April 2022). However, we have to be aware that ground based data in a city may differ from columnar observations due to the different altitude range and local chemical processes.*

*The data are illustrated in Figure 4 below. I averaged the available data for the respective city and for the noon time (12-14 local time). While the summer time maxima for Berlin and Athens are almost the same the winter time minima are higher in Athens. A similar effect is not seen in the satellite data. But both the ground based and satellite data show a stronger variability during the summer months for Berlin compared to Athens. The low ozone concentrations in Athens in summer 2020 will not be discussed here.*

[Figure]

*Figure 4: Ozone concentrations [µg/m³] in Berlin (top)
and Athens (bottom) for the years 2018, 2019, 2020.The
figure will not be included in the manuscript*

Lines 322-324: "When the different .  data product is reasonable"  This needs rephrasing. You probably mean "the data agreement is reasonable".
*Corrected*:
*The agreement between the data products is reasonable*

Lines 330-333. This paragraph on future plans looks strange in the middle on conclusions. The second sentence is not clear, in particular, why long-term dataset is needed for evaluation of COVID-19 lock down measures.
*From an atmospheric point of view the COVID lock down measures in 2020 can be seen as "large scale emission reduction experiment" especially the tropospheric ozone precursor $NO_x$ was reduced. This might cause changes in the tropospheric ozone as well. However, also the meteorology affects the tropospheric ozone burden. So the normal non lock down ozone column varies depending not only on the $NO_x$ emissions. Because of that the ozone column can not be directly compared from 2018/2019 to the 2020 data. A longer time series helps to estimate the typical variability, which might be in the order of the reduction caused by the COVID lock down. added,*
*the conclusion is reorganised*

TECHNICAL CORRECTIONS

Line 1 Misprint in TROPOMI
*corrected*

Line 3. Microwave Limb Sounder (capitalize first letters)
*corrected*

Line 7  "S5P_O3_TCL"  is not needed in the abstract

*deleted, not needed at all.*

Line 83:  TROPOMI acronym should be explained above in the text.
*TROPOspheric Monitoring Instrument added in line 26*

Line 88: Remove "Clouds as Layers" before  "Loyola et al., 2018".
*removed*

Line 94 UPAS version 1.x ? (Should be a number instead of "x")
*The climatology was used for all subversion of UPAS 1 the ".x" was removed*

Line 95: misprint in "latest"
*removed, updated to version 2.1.3 meanwhile updates are available*

Line 123:  alternate -> alternative
*corrected*

Line 184:  (0, 3, 6, ?, 21 UTC) -> (0, 3, 6, ..., 21 UTC)
*corrected*

Line 248 Figure7
*corrected*

Line 256: date -> data
*corrected*

Line 258:100km -> 100 km
*corrected for all units*

Line 275    found, which ...
*corrected*

Figure 14 caption, misprint in "tropospheric"
*I can't find this misprint in: "Figure 14. Formaldehyde (top) and tropospheric ozone (bottom) over the United States observed in July 2018. Formaldehyde is a tropospheric ozone precursors."*

*Moxim, W., Levy II, H., 2000. A model analysis of the tropical South Atlantic ocean tropospheric ozone maximum: the interaction of transport and chemistry. Journal of Geophysical Research 105 (D13), 17,393e17,415. doi:10.1029/2000JD900175.*

This is a very nice tropospheric column ozone product and the fine horizontal resolution is amazing.

Regarding the case study of high ozone values above the southeastern USA, these enhancements are almost certainly due to high ozone levels in the upper troposphere, due to photochemical ozone production from lightning NOx. The papers listed below show the results from targeted ozonesonde campaigns that were designed to study this upper tropospheric phenomenon. To confirm that the TROPOMI ozone enhancements are in the upper troposphere you could check the ozonesondes from Huntsville, Alabama and the IAGOS aircraft profiles.

Ozonesonde profiles from Huntsville, Alabama can be found here:

https://gml.noaa.gov/aftp/data/ozwv/Ozonesonde/Huntsville,%20Alabama/100%20Meter%20Average%20Files/

IAGOS ozone profiles above the southeastern USA (Atlanta, Dallas and Houston) can be found here:
http://iagos-data.fr/#TimeseriesPlace:

*Thank you Owen for pointing this out. We tend to believe that the tropospheric column always varies in the boundary layer but of cause this is not true. I included the available soundings from Huntsville and Trinidad head in the comparison plots.*
*The IAGOS profiles, however are not yet included.*

*Cooper, O. R., S. Eckhardt, J. H. Crawford, C. C. Brown, R. C. Cohen, T. H. Bertram, P. Wooldridge, A. Perring, W. H. Brune, X. Ren, D. Brunner, and S. L. Baughcum (2009), Summertime buildup and decay of lightning NOx and aged thunderstorm outflow above North America, J. Geophys Res., 114, D01101, doi:10.1029/2008JD010293.*

*Cooper, O. R., M. Trainer, A. M. Thompson, S. J. Oltmans, D. W. Tarasick, J. C. Witte, A. Stohl, S. Eckhardt, J. Lelieveld, M. J. Newchurch, B. J. Johnson, R. W. Portmann, L. Kalnajs, M. K. Dubey, T. Leblanc, I. S. McDermid, G. Forbes, D. Wolfe, T. Carey-Smith, G. A. Morris, B. Lefer, B. Rappenglück, E. Joseph, F. Schmidlin, J. Meagher, F. C. Fehsenfeld, T. J. Keating, R. A. Van Curen and K. Minschwaner (2007), Evidence for a recurring eastern North America upper tropospheric ozone maximum during summer, J. Geophys. Res., 112, D23304, doi:10.1029/2007JD008710.*

*Cooper, O. R., A. Stohl, M. Trainer, A. Thompson, J. C. Witte, S. J. Oltmans, G. Morris, K. E. Pickering, J. H. Crawford, G. Chen, R. C. Cohen, T. H. Bertram, P. Wooldridge, A. Perring, W. H. Brune, J. Merrill, J. L. Moody, D. Tarasick, P. Nédélec, G. Forbes, M. J. Newchurch, F. J. Schmidlin, B. J. Johnson, S. Turquety, S. L. Baughcum, X. Ren, F. C. Fehsenfeld, J. F. Meagher, N. Spichtinger, C. C. Brown, S. A. McKeen, I. S. McDermid and T. Leblanc (2006), Large upper tropospheric ozone enhancements above mid-latitude North America during summer: In situ evidence from the IONS and MOZAIC ozone monitoring network, J. Geophys. Res., 111, D24S05, doi:10.1029/2006JD007306.*

---

## Author Response (AR2)

Minor comments
Line numbers are as in the revised manuscript.

-Please correct punctuations on lines 106 and 265.
D*one*

-L 135: "Within this project" -> "In this work"?
*Done*

-L.138: "first the PV, PT tropopause" Acronyms should be explained
*Done*

-L. 300-302: "The global mean difference between these two tropospheric ozone data equals 1.82 DU and might therefore explain at least half of the differences between our tropospheric ozone data set and OPMS-MERRA-2." As seen in Figure 7 (top), the differences in ozone due to different tropopause definitions in NH is ~2-5 DU, while the difference S5P-BASCOE minus OMPS-MERRA-2 (Figure 6, middle) is ~7-20 DU, which is much larger. Therefore, please either (i) replace "at least half" with "partly" or (ii) subtract the estimated difference due to different tropopause definitions from" S5P-BASCOE minus OMPS-MERRA-2" (add a subplot in Figure 6)

*A subplot was added (S5-BASCOE(2.5PVU) - OMPS-MERRA-2) to figure 7. It is the same as suggested by the reviewer (middle of figure6 - upper part of figure 7). Based on this we correct our previous statement from "at least half" to "to a large extend".*

-Related to shifting to the ocean and not observing strong enhancements over Africa (L 363- 367), you can also mention that this is a combined effect of TROPOMI low sensitivity near the ground and wind advection of both ozone and its precursors towards the west in the middle troposphere, as found by simulations using a chemistry-transport model (Sofieva et al., 2022).

*Thank you for pointing this out. The study by Sofieva et al. (2022) confirms the previous study by Maxim and Levy (2000) with respect to the transport of polluted air masses to the Atlantic and the enhanced concentration in the middle troposphere. TROPOMI and similar UV instruments are more sensitive to the trace gases in the middle troposphere compared to the ground level.*
*included:*
*It is remarkable that the ozone column over the African Continent is lower compared to the Atlantic ocean. The low sensitivity of TROPOMI to ozone in the lower troposphere might cause an underestimation if the ozone concentration is enhanced close to ground.*
*Tropospheric ozone over the tropical Atlantic ...*
Sofieva et al. (2022) included chemical transport models in their study and confirmed the enhanced columns over the Southern Atlantic in the middle troposphere. They also found low tropospheric columns over the African continent that can be attributed to the low sensitivity of UV nadir viewing satellites for boundary layer trace gases.

Reference:
Sofieva, V. F., Hänninen, R., Sofiev, M., Szeląg, M., Lee, H. S., Tamminen, J. and Retscher, C.: Synergy of Using Nadir and Limb Instruments for Tropospheric Ozone Monitoring (SUNLIT), Atmos. Meas. Tech., 15(10), 3193–3212, doi:10.5194/amt-15-3193-2022, 2022.